# Very Large-Scale Multi-Agent Simulation with LLM-Powered Agents

## Abstract

Recent advances in large language models (LLMs) have opened new avenues for applying multi-agent systems in very large-scale simulations. However, there remain several challenges when conducting multi-agent simulations with existing platforms, such as limited scalability and low efficiency, unsatisfied agent diversity, and effort-intensive management processes. To address these challenges, we develop several new features and components based on a user-friendly multi-agent platform, enhancing its convenience and flexibility for supporting very large-scale multi-agent simulations. Specifically, we propose an actor-based distributed mechanism as the underlying technological infrastructure towards great scalability and high efficiency, and provide flexible environment support for simulating various real-world scenarios, which enables parallel execution of multiple agents, automatic workflow conversion for distributed deployment, and both inter-agent and agent-environment interactions. Moreover, we develop an easy-to-use configurable tool and an automatic background generation pipeline, simplifying the process of creating agents with diverse yet detailed background settings. Last but not least, we provide a web-based interface for conveniently monitoring and managing a large number of agents that might deploy across multiple devices. We conduct a comprehensive simulation to demonstrate the effectiveness of these proposed enhancements, and provide detailed observations and insightful discussions to highlight the great potential of applying multi-agent systems in large-scale simulations.

## 1 Introduction

Large language models (LLMs) (OpenAI, 2023; ANTHROP, 2024; MistralAI, 2024) demonstrate remarkable abilities in understanding, generating, and interacting with human language. Recent advancements in LLMs have sparked a revolution in natural language processing and related fields, paving the way for novel applications that were previously inconceivable. Building on the capabilities of LLMs, there is growing interest in the development of intelligent agents, which have the potential to redefine the landscape of simulations (Matsumoto et al., 2024; Sorokovikova et al., 2024; Sreedhar & Chilton, 2024; Yue et al., 2024) towards more interactive, adaptive, and realistic, while requiring substantially fewer human efforts.

Recently, several platforms (Hong et al., 2024b; Wu et al., 2023; Team, 2023) have been proposed to streamline the development of multi-agent systems, providing some fundamental functionalities including unified LLM services, various tools, and advanced reasoning algorithms. Despite significant progress, we identify several challenges in conducting simulations with multi-agent platforms, particularly when the number of agents becomes extremely large:

(i) **Scalability and Efficiency Limitations**. The scale of involved agents can be critical since simulations at a small scale run the risk of inaccurately representing real-world complexities, making simulations less realistic and reliable (Macal & North, 2010; Macal, 2016). However, increasing the scale of agents brings challenges to the simulation platform in terms of scalability and efficiency. It is non-trivial to efficiently organize agents to execute their tasks and communications following an appropriate order, with the aim of reducing the running time while ensuring accurate results.

(ii) **Unsatisfied Population Distributions and Agent Diversity**. For a large-scale simulation, it is essential that the involved agents exhibit diverse behaviors while generally following a specific population distribution (Gao et al., 2023; Ren et al., 2024). Assigning agents with simple backgrounds may result in a significant number of highly homogeneous agents, making it difficult to derive meaningful insights. The existing studies rarely consider how to specify population distributions of agents from different perspectives, such as age, education, occupation, etc.

(iii) **Difficult Management Processes**. As the scale of agents increases, it becomes rather effort-intensive to manage the simulations, including initialization, execution, and termination of a large number of agents spread across multiple devices, as well as monitoring their status, behaviors, and interactions (Mou et al., 2024). Such difficulties in managing

make it challenging to promptly identify valuable group-level and individual-level behaviors, which can further hinder the discovery of critical insights for optimizing simulations and advancing research.

To tackle these challenges, we adopt a user-friendly multi-agent platform, named AgentScope (Gao et al., 2024), as the foundation framework to provide the basic functionalities, and further develop several new features and components upon it to improve its usability, convenience, and flexibility for supporting very large-scale multi-agent simulations.

Specifically, we propose a distributed mechanism based on the actor model (Agha, 1985), featuring agent-level parallel execution and automatic workflow conversion to provide great scalability and high efficiency for multi-agent-based simulations. The proposed actor-based distributed mechanism enables us to further expand the scale of agents in the simulation with a limited number of devices, and provides linear benefit on running time from the addition of devices. Users can convert the simulations from a centralized workflow into a distributed one without further modifications except for adding a `to_dist` function. Besides, we support both inter-agent and agent-environment interactions in the simulations. Agents can communicate with each other, and can query some shared states in the environment, respond to their changes, and make modifications as needed. We enable user-defined functions within the environment to support flexible extension of various states and their corresponding query and modification behaviors.

To satisfy the requirements of population distribution and agent diversity, we provide a configuration tool and an automatic background generation pipeline. Users only need to simply specify the distributions of the population from several aspects, a large number of agents with detailed and diverse characteristics can be effortlessly generated accordingly. These agents can be managed and monitored conveniently through Agent-Manager, a proposed module for simplifying the organization and observation process of large-scale agent-based simulations. Using a web-based visual interface, Agent-Manager provides a comprehensive overview of all agents across multiple devices, allowing users to efficiently configure, launch, and terminate these agents.

With such an agent-based simulation platform, we conduct a comprehensive simulation on the classic "guess $\frac{2}{3}$ of the average" game (Nagel, 1995; Camerer et al., 2004) to demonstrate the improvements and advances brought by the infrastructure introduced above. Firstly, we conduct agent-based simulations involving 1 million agents using only 4 devices, showing the scalability and efficiency of the platform. Then, we incorporate agents using different LLMs of different sizes, equipped with different prompts and diverse background settings, resulting in various and realistic behaviors in the simulations. We provide detailed observations on both collective and individual behaviors, drawing meaningful and valuable insights from a series of simulation experiments, along with further discussions on helpful tips and open questions. These experimental results confirm the feasibility and great potential of conducting large-scale agent-based simulations with LLM-powered agents.

## 2 RELATED WORKS

**LLM-Powered Agent Platforms**  A significant number of agent platforms have been developed to integrate LLMs into real-world applications and assist humans in problem-solving (Hong et al., 2024b; Li et al., 2023b; Significant-Gravitas, 2023; Team, 2023; Chen et al., 2024b; Gao et al., 2024). These platforms can be categorized into single-agent platforms and multi-agent platforms. The single-agent platforms include AutoGPT (Significant-Gravitas, 2023), LangChain (langchain ai, 2024a), ModelScope-Agent (Li et al., 2023a), and Transformers Agents (Wolf et al., 2020), which are proposed to resolve practical tasks using LLMs. On the other hand, multi-agent platforms like MetaGPT (Hong et al., 2024b), Auto-Gen (Wu et al., 2023), CAMEL (Li et al., 2023b), and LangSmith (langchain ai, 2024b) employ multi-agent collaboration to tackle more complex challenges, including software programming (Hong et al., 2024b; Qian et al., 2024), data science (Hong et al., 2024a), social simulation (Park et al., 2023), game-playing (Chen et al., 2024a), etc. Although remarkable progress has been made, applications built on these platforms can currently be limited in the scale of agents and suffer from low efficiency, hindering their potential for large-scale simulations.

**Agent-Based Simulation Frameworks**  Recently, agent-based simulation has become an attractive topic in the research community (Gürcan, 2024; Sorokovikova et al., 2024; Sreedhar & Chilton, 2024; Ye & Gao, 2024; Team et al., 2024; Park et al., 2023). Previous studies have explored the integration of LLMs in various fields, including education (Yue et al., 2024), economic (Matsumoto et al., 2024), societal study (Park et al., 2023; Ye & Gao, 2024; Gao et al., 2023; Ren et al., 2024), transportation (Jin et al., 2023), healthcare (Zhang et al., 2023), etc. Researchers have built up several LLM-based or agent-based simulation frameworks. For instance, Vidur (Agrawal et al., 2024) is a simulation framework that focuses on providing high-throughput LLM services, SOTOPIA (Zhou et al., 2024) provides an environment to simulate various social scenarios and evaluates the social intelligence of agents, Cheng et al. (2023) proposes to evaluate the level of caricature, and Ren et al. (2024) designs a framework to simulate the behaviors of web search users. However, these existing frameworks are domain-specific and lack flexibility and extensibility, making it challenging for users to conduct large-scale agent-based simulations for a wide variety of applications.

Figure 1: The example of the automatic conversation of a centralized workflow with four agents running sequentially (on the left) into a distributed workflow with agents running in parallel (on the right). Users only need to add a `to_dist` function during the initialization phase, without any further modifications required.

## 3 INFRASTRUCTURE

To provide the basic functionalities required for conducting agent-based simulations, we adopt AgentScope, a user-friendly multi-agent platform, as our foundation framework. We further develop several new features and components, making it more convenient and feasible to support very large-scale simulations involving multiple agents. In the following subsections, we elaborate on the details of these proposed enhancements.

### 3.1 ACTOR-BASED DISTRIBUTED MECHANISM

The actor model is a mathematical model of concurrent computation, where each actor acts as a basic computing unit, receives messages, and computes independently (Agha, 1985). Based on the actor model, we design a distributed mechanism to provide *agent-level parallel execution* for achieving high efficiency and great scalability for agent-based simulation, and to support *automatic workflow conversion* for migrating the centralized orchestrated workflow into distributed scenarios effortlessly.

**Agent-Level Parallel Execution** In a multi-agent simulation, the interactions between agents follow an atomized pattern, where interactions occur within small isolated cliques (Matsumoto et al., 2024; Sorokovikova et al., 2024). Such a pattern holds significant potential for parallelization, leading to substantial gains in efficiency.

With the actor model, agents that do not rely on the outputs of others or whose dependencies have all been satisfied, can be executed in parallel for enhancing efficiency. As agents complete their executions and produce results that may be needed by others, some previously waiting agents become active and initiate their executions. An example of the proposed agent-level parallel execution can be found in Appendix B.1.

We propose two multi-process modes for various simulation scenarios, supporting both one-to-one and many-to-one relationships between agents and processes. Communication across processes and devices utilizes Remote Procedure Call, and we provide implementations in both Python and C++. The one-to-one multi-process mode, where each agent runs in a separate process, is well-suited for agents performing computation-intensive tasks, ensuring that each agent has sufficient computational resources and that the parallelization would not be hindered by Global Interpreter Lock (GIL). For the many-to-one multi-process mode where multiple agents run within a single process, agents within a single process share CPU cores, communication ports, and global variables. Such a mode is well-suited for situations where agents experience I/O wait times or are awaiting responses from remote APIs, allowing the CPUs to leverage the time-sharing mechanism to maximize resource utilization.

In this way, the proposed framework achieves higher resource utilization in parallelism compared to existing ones (Wu et al., 2023; Hong et al., 2024b) that rely on asynchronous I/O in Python, which can be constrained by GIL. Furthermore, it also makes a significant advancement over existing actor-based distributed frameworks, such as Ray (Moritz et al., 2017), which allocate a new worker process for each actor, resulting in wasted computational resources when applying for large-scale agent-based simulations.

**Automatic Workflow Conversion** To ease the distributed deployment of large-scale simulations, we provide an easy-to-use function called `to_dist` to convert a centralized workflow into a distributed one effortlessly. The conversion involves two automatic stages. In the first stage, each agent in the center (*i.e.*, the main procedure) is distributed to a

specified device, with a *proxy* left in the center as a substitute. This proxy can be utilized for users to orchestrate the workflow, maintaining parity with the centralized mode, and is responsible for automatically forwarding messages to the corresponding distributed agents. In the second stage, *placeholders* are introduced to ensure that calculations performed by the distributed agents do not block the workflow execution in the center. When a proxy receives a message, it immediately returns a placeholder and forwards the message to its corresponding distributed agent for processing. For a distributed agent, when receiving a message, it first checks if the message contains placeholders to decide whether it needs to request and wait for the results from the agents indicated on the placeholders. This mechanism allows the main procedure to continue executing without waiting for the results from the distributed agents.

The example in Fig. 1 shows how a centralized workflow can be automatically converted to a distributed one, without any further modifications except for adding `to_dist`. Firstly, all agents are transformed into distributed agents, with corresponding proxies left at the center. Then, the center can continue its execution until it needs to print `msg_b` (*i.e.*, line 15 of the code), thanks to the usage of placeholders that replace messages `msg_a/b/c/d` as responses. Once receiving `msg_a` from agent-A, agent-B and agent-C run in parallel to produce `msg_b` and `msg_c`, respectively, which would be finally sent back to the center for printing out.

### 3.2 AGENT-ENVIRONMENT INTERACTIONS

Agent-environment interactions are also crucial alongside the above inter-agent communication in agent-based simulations, which indicates that agents can access the shared states in the environment, respond to their changes, and make modifications as needed.

For large-scale simulations, the environment is expected to meet the following requirements: (a) *High concurrency access*. Given a large number of agents, the environment should support high concurrency access, especially when agents need to frequently check and interact with the environment's states. (b) *Diverse states*. Different simulation scenarios may need different states to be maintained in the environment (Yang et al., 2024b; Xi et al., 2024). For example, in a chat room simulation, the states might include participants and conversation history, while in a maze simulation, the states contain the locations of agents and other interactable items. These diverse states pose challenges regarding the flexibility and extensibility of the environment module. (c) *Bi-directional interaction*. The interactions between agents and the environment should be bidirectional (Beer, 1995), indicating that agents can query or modify the states in the environment, and the environment can also actively send notifications or requests to agents. For example, a chatroom can notify specific participants when they are mentioned or called. (d) *Multiple environments*. Some simulation scenarios might need to have multiple environments concurrently. For example, in a social simulation where agents are divided into subgroups, agents within a subgroup may share one dedicated environment for collaborative tasks or information synchronization without interference from other groups. Multiple environments allow for more tailored interactions and state management specific to each subgroup's unique objectives.

To satisfy all the aforementioned requirements, we abstract the environment as a special type of agent with rich functionalities, which allows the environment to maintain shared variables as states and to communicate with other agents through Remote Procedure Calls for ensuring high concurrency. Meanwhile, we enable user-defined functions within the environment to support flexible extension of various states and their corresponding query and modification behaviors. For bi-directional interactions, we introduce listeners, which can be attached to user-defined functions in the environment, automatically triggering the sending of messages from the environment to agents when specific conditions are met. Furthermore, we support the nesting of environments and agents to allow multiple environments within a single simulation. An example of a multi-environment designed for group-wise information synchronization is shown in Appendix B.2, and a use case can be found in Sec. 4.5.

### 3.3 HETEROGENEOUS CONFIGURATIONS

In a simulation, agents are expected to act as humans with diverse backgrounds, including different ages, genders, careers, nationalities, education, experiences, etc. An intuitive approach is to add these background settings of agents in their system prompts, providing guidance for agents on the roles to play and actions to take. However, for large-scale simulations, providing diverse, heterogeneous, and reasonable background settings for agents can be laborious and time-consuming, especially when precise control of different population distributions is required in certain simulations. This problem motivates us to provide easy-to-use tools to assist users in effortlessly setting up large-scale agents with diverse background settings.

**Configurable Tool** Specifically, users can begin by defining the total population of the simulation, and then specify the distributions of the population from various perspectives. We provide some widely-used distribution templates for convenient usage, from the aspects of age, gender, occupation, nationality, and education. Besides, the proposed

configurable tool allows for easy extension of new aspects, enhancing its flexibility to meet diverse requirements. An example configuration file is shown in Appendix B.3 for a group of people with different educational levels, in which the proportions of its different components can specify a distribution.

**Automatic Background Generation Pipeline**  After configurations have been provided via the above tool, more detailed and heterogeneous background settings can be automatically generated to instantiate the agents. Specifically, when users start a simulation, we draw specific values from the distributions based on the configurations, convert them into a JSON format, and fill them into a meta prompt to produce the completed instructions for background generation tasks. These instructions are utilized by LLMs to generate heterogeneous background settings. To introduce more diversity, the generation process involves adjusting the random seed and the temperature used by LLMs. Several examples of the generated background settings can be found in Sec. 4.4, along with the results and analysis of the simulations involving diverse agents.

### 3.4 MANAGEMENT FOR LARGE-SCALE AGENTS

In a simulation, users need to manage and monitor a large number of agents distributed across different devices, which might become intractable to handle manually as the scale and complexity of the simulation increase. To tackle this, we incorporate advanced forms of agent management and monitoring, named Agent-Manager. Specifically, when users start a simulation, servers are first launched on all the remote devices, which provide resident services to remotely create, monitor, and stop distributed agents. These servers are responsible for managing the lifecycle of distributed agents and synchronizing their information to a web-based visual interface, which is provided for a comprehensive overview of all registered servers and all deployed agents on different devices, as shown in Appendix B.4.

The Agent-Manager also simplifies the management and monitoring processes for conducting multiple simulations. Since the servers can be reused in different simulations, users don't need to restart the distributed servers between two simulations. Users can efficiently configure, launch, and terminate servers and agents during the simulations as needed. With such a design, we streamline the management process by focusing on servers rather than individual agents, thereby improving the efficiency and effectiveness of managing large-scale agent systems.

### 4 EXPERIMENTS

In this section, we conduct large-scale simulations to show the improvements and advances brought by the proposed infrastructure and components. Meanwhile, we provide detailed observations and in-depth discussions on the agents' collective and individual behaviors, drawing valuable insights.

### 4.1 SETTINGS

We set up a large number of agents to participate in the classic game *guess the $\frac{2}{3}$ of the average*, where each agent reports a real number between 0 and 100 and the agent who reports a number closest to $\frac{2}{3}$ of the average of all the reported numbers wins the game. In this game, intuitively the highest possible average is 100. Therefore, for winning the game, agents tend to report a number no larger than $100 \times \frac{2}{3} = 66\frac{2}{3}$. Once all agents adopt this strategy, $66\frac{2}{3}$ becomes the new highest possible average and thus they tend to report a number no larger than $66\frac{2}{3} \times \frac{2}{3} = 44\frac{4}{9}$. This process continues until the average becomes 0 and all agents report 0, indicating that the game has reached its Nash equilibrium. However, considering that agents may not always be rational, those agents who report 0 cannot always win the game since the average does not converge to 0 immediately. Agents should carefully take into account the possible actions of others before reporting their numbers. Meanwhile, agents can adjust their strategies in a multi-round game according to the average reported numbers in previous rounds. **Note that all the experiments in this section follow the aforementioned settings.**

**Devices & LLMs**  The experiments are conducted on a cluster containing multiple devices, each equipped with 8 A100-80G GPUs, a 64-core CPU, and 1 TB of memory. We adopt vLLM (Kwon et al., 2023) as the LLM inference engine to handle highly concurrent LLM service requests. We utilize six powerful and popular open-source LLMs of different sizes, including Llama3-8B / Llama3-70B (Meta, 2024), Qwen2-7B / Qwen2-72B (Yang et al., 2024a), and MistralAI-8x7B / MistralAI-8x22B (MistralAI, 2024). We adopt their instruction versions due to their enhanced ability to follow instructions. The details of the adopted LLMs and implementations can be found in Appendix C.

**System Prompts**  We provide system prompts for agents to guide them in defining their dialogue style, background knowledge, task requirements, and so on. To be more specific, for playing this game, the system prompt incorporates the game rules and response formats, as illustrated in Prompt 1. Besides, we can include further behavioral guidance

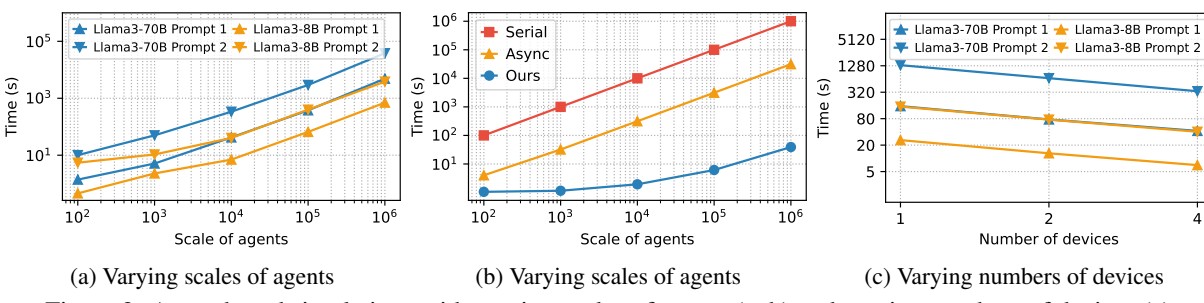

| Prompt 1 | Prompt 2 |
|---|---|
| You are playing a multiplayer game.

# Game Rule
1. Each player reports a real number between 0 and 100, inclusive.
2. The winner will be the player whose number is the closest to 2/3 of the average of all reported numbers.

Directly report your number without additional information. | You are playing a multiplayer game.

# Game Rule
1. Each player reports a real number between 0 and 100, inclusive.
2. The winner will be the player whose number is the closest to 2/3 of the average of all reported numbers.

*Think step by step and then report your number.* |

  (a) Varying scales of agents   (b) Varying scales of agents   (c) Varying numbers of devices

Figure 2: Agent-based simulations with varying scales of agents (a, b) and varying numbers of devices (c).

in the system prompts to encourage behaviors that more closely resemble those of real human beings. For example, inspired by "chain-of-thought" studies (Wei et al., 2022; Wang et al., 2023), we ask agents to think step by step before reporting their numbers, producing the system prompt shown in Prompt 2.

## 4.2 SCALABILITY AND EFFICIENCY

First of all, we conduct a series of experiments to show the scalability and efficiency of the agent-based simulations supported by the proposed actor-based distributed mechanism (see Sec. 3.1). Specifically, we illustrate how the overall simulation running time changes as the number of participating agents grows when using LLMs of different sizes, including Llama3-8B and Llama3-70B. In addition to the model sizes, the system prompt provided to agents is also a factor that can influence the running time, since some prompts (*e.g.*, Prompt 2) may encourage agents to generate longer responses and thereby lead to longer response time. From the experimental results shown in Fig. 2, we can obtain the following observations and insights.

(i) **We support an agent-based simulation involving 1 million agents, which can be completed in 12 minutes using 4 devices.** In Fig. 2a, we fix the device number to 4 and record the simulation running time as the number of agents grows from 100 to 1M. It can be observed that the simulation involving 1 million agents finishes in 12 minutes when using Llama3-8B with Prompt 1, while it takes 85 minutes if we choose Prompt 2, as the number of averaged response tokens grows by more than 150-fold[1]. For the heaviest inference workload, *i.e.*, when agents adopt Llama3-70B and Prompt 2, it takes around 10.6 hours to complete the simulation.

(ii) **The proposed actor-based distributed mechanism significantly improves the efficiency of large-scale agent-based simulations.** To better demonstrate the improvements brought by the proposed actor-based distributed mechanism, we adopt a dummy model request (*i.e.*, agents sleep for 1 second and generate random numbers rather than posting the requests) in the simulation to remove the impact of the LLM inference speed. The experimental results summarized in Fig. 2b show that, completing an agent-based simulation with the proposed actor-based distributed mechanism involving 1 million agents only takes 40 seconds, whereas simulations using serial execution or asynchronous mode in Python (adopted by existing works (Wu et al., 2023; Hong et al., 2024b)) require around 12 days and 8.6 hours, respectively.

(iii) **Increasing the number of devices can proportionally reduce the simulation running time.** As shown in Fig. 2c, we maintain the number of agents at 10,000 and vary the number of devices used in the simulation. For Llama3-70B with Prompt 2, the simulation running time decreases from 22 minutes to 5.6 minutes as the number of devices increases from 1 to 4. Such a phenomenon can be attributed to a reduction in the number of agents served within one device, which demonstrates the horizontal scalability of the proposed framework.

---

[1]The number of response tokens when using different LLMs and system prompts are summarized in Appendix D.

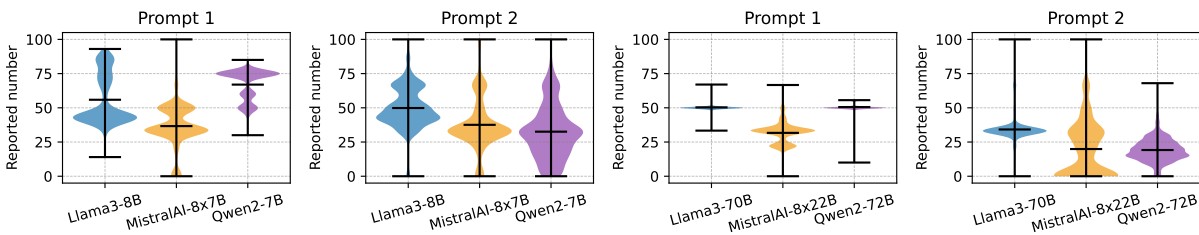

Figure 3: The distributions of numbers reported by agents with different LLMs and system prompts.

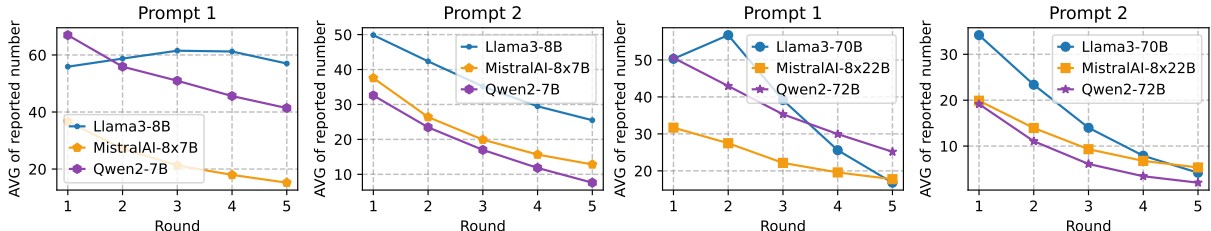

Figure 4: The average of reported numbers in a multi-round game.

In summary, the proposed actor-based distributed mechanism enhances the efficiency of very large-scale simulations, and offers great scalability by allowing users to expand the scale of agents from the addition of devices.

### 4.3 SIMULATION RESULTS AND ANALYSIS

In this subsection, we add some detailed information with six LLMs and two system prompts. We summarize the experimental results in Fig. 3, from which we derive the insights below.

From the comparisons in the figures, we observe that when utilizing a basic system prompt Prompt 1 for most LLMs, agents generally tend to report numbers around 50. However, it is worth noting that agents with MistralAI-8×7B and MistralAI-8×22B, report smaller numbers (36.63 and 31.69 in average, respectively) than other agents. These results indicate that without providing specific instructions in the system prompt, the performance of agents can be different due to the LLMs they adopt, influenced by factors such as model sizes and model architectures.

When we change the system prompt to a chain-of-thought prompt (*i.e.*, Prompt 2), the reported numbers of agents move forward zero markedly, with $\frac{2}{3}$ of the average being much smaller than those using Prompt 1, *e.g.*, decreasing from 33.70 to 12.76 when using Qwen2-72B. Meanwhile, we notice that more than 30% of agents using MistralAI-8×22B report around 0, leading to the Nash equilibrium of this game. These observations in the simulations demonstrate the effectiveness and importance of providing suitable system prompts for guiding agents to perform the thought processes.

Moving forward, we expand the simulation to multiple rounds. We inform agents of the winner number in the previous round at the beginning of each round except the first one, and request each agent to report a number at each round. Informing these winner numbers to agents enables them to adjust their strategies accordingly. Such a process is implemented based on the agent-environment interaction mechanism (see Sec. 3.2), with the winning numbers being set as shared states in the environment.

The experimental results are demonstrated in Fig. 4. From these results we can observe that, as the game progresses from round to round, the reported numbers of agents gradually converge to 0, indicating that agents have a good understanding of this game and are capable of considering other agents' behaviors and making rational decisions. Similarly, using the chain-of-thought prompt can accelerate the game to reach its Nash equilibrium. For example, in the fifth round, the average reported number of agents using Qwen2-72B with Prompt 2 is 2.02, which is significantly smaller than those using Prompt 1, reporting 25.16.

It is worth noting that these experimental results are consistent with previous studies (Nagel, 1995; Camerer et al., 2004) in social simulation, which confirms the reliability and potential of multi-agent-based simulations.

For the above experiments, we provide more detailed results (*e.g.*, distributions and statistics of the reported numbers), individual-level observations, and case studies in Appendix E.1 and E.2. Besides, to further explore the impact of behavioral guidance on agents, we incorporate more detailed instructions tailored for this game in the system prompts, and summarize the experimental results in Appendix E.3.

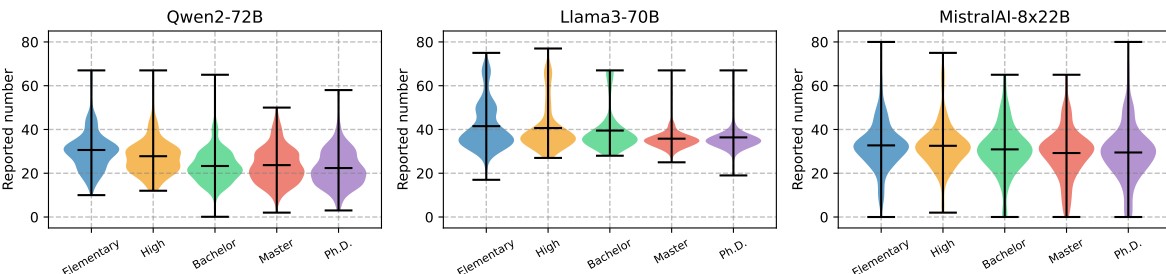

Figure 5: The distributions of numbers reported by agents characterized by different educational levels.

## 4.4 DIVERSE BACKGROUND SETTINGS

The diversity of agents is a crucial factor in agent-based simulations. In Sec. 3.3, we introduce a configurable tool and background generation pipeline designed to automatically instantiate agents with varied background settings. Utilizing these components, we conduct simulation experiments that incorporate agents with diverse background settings.

Specifically, we divide the agents into several groups, each of which consists of 200 agents. We manually provide a basic configuration for each group, and utilize LLMs to generate a detailed description for agents within the group. These generated background settings are added to the system prompts and labeled as "character background". We set up a simulation experiment involving agents with different educational levels, in which we assign five different educational levels to agents, including elementary school, high school, bachelor, master, and Ph.D.

The simulation results are illustrated in Fig. 5, from which we can derive the following insights. In general, the higher the educational level of agents, the lower the average reported numbers, indicating more rational behaviors in this game. For example, when using Qwen2-72B, the average numbers reported by agents with a Ph.D. education are much lower than those reported by agents characterized as primary school students. Meanwhile, from the individual-level behaviors, we notice that agents can effectively perform reasoning processes and make corresponding decisions according to the assigned roles. Besides, different LLMs demonstrate varying sensitivities to educational levels in the background settings. For example, MistralAI-8×22B has the least sensitivity, with the largest difference in average reported numbers is 3.49 (32.73 v.s. 29.24), while that of Llama3-70B and Qwen2-72B is 5.72 and 8.24, respectively.

The prompt used to generate background settings, and case studies of the generated results and agent behaviors can be found in Appendix F.1 and F.2. We also conduct a simulation experiment involving agents with different occupations. The experimental results shown in Appendix F.3 further confirm that agents' considerations and actions are consistent with their respective occupations.

## 4.5 MIXTURE OF LLMS

In this subsection, we conduct a simulation experiment involving agents employing a mixture of LLMs. Specifically, we configure agents employing Llama3-70B, MistralAI-8×22B, and Qwen2-72B, with 500 agents assigned to each LLM. We conduct both individual-level simulations, where each agent plays the game independently, and group-level simulations, where agents using the same LLMs form a group.

**Individual-Level Simulations** The simulation results are illustrated in Fig. 6. At the first round of the game, we observe that agents with Llama3-70B exhibit similar behaviors, tending to report numbers around 33, while agents with MistralAI-8×22B consistently report 0. On the other hand, agents with Qwen2-72B exhibit more diverse behaviors, reporting a wider range of numbers, with most of them falling between 0 and 50. These behaviors can be attributed to the preferences of LLMs, which may be related to their architectures, training corpus, etc.

As the game progresses round by round, agents are informed of the winning number from the previous round and adjust their strategies accordingly. As shown in Fig. 6, the majority of agents report numbers close to the winning number in the previous round, with approximately 59.7% reporting numbers smaller than the previous winning number. We present a typical response in Appendix G, where an agent adopts a conservative strategy and chooses a number slightly smaller than the winner number 15.90.

In the pie chart of Fig. 6, we show the winners of each round in the simulation, grouped by their employed LLMs. To reduce the randomness in the simulation, we regard those agents whose reported numbers fall within the range of $\pm 0.5$ from $\frac{2}{3}$ of the average as the winners. The figure shows that in the first and fifth rounds, agents equipped with Qwen2-72B outperform other agents, while agents equipped with MistralAI-8×22B emerge as winners in the second,

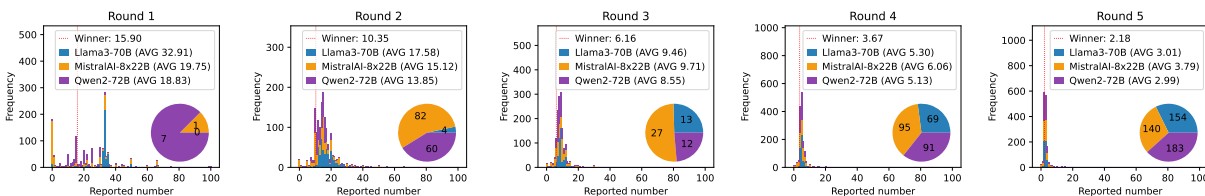

Figure 6: Individual-level simulations involving agents with a mixture of LLMs.

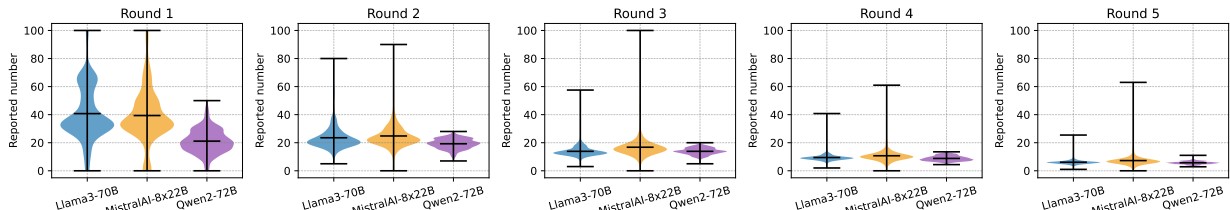

Figure 7: Group-level simulations involving agents with a mixture of LLMs.

third, and fourth rounds. Notably, almost all agents tend to report numbers near $0$ in the final round, indicating agents can perform reasonable considerations and behaviors to promote this game approach to its Nash equilibrium.

**Group-Level Simulations**  In a group-level simulation, agents are divided into three groups, and each agent reports a number. The average number of agents within each group is regarded as the reported number of this group. Finally, the group that reports a number that is closest to the $\frac{2}{3}$ of the average among the groups' reported numbers wins the game.

Meanwhile, we modify the information announced between different rounds in a multi-round game. Starting from the second round, in addition to the winning number of the previous round, all agents are informed of the reported numbers from all three groups in the previous round. This provides additional guidance for agents to adjust their strategies. Such a group-wise synchronization is implemented based on the agent-environment interaction mechanism (see Sec. 3.2), allowing agents within the same group to share an interactive environment for synchronization.

The simulation results are shown in Fig. 7. From the figures, it can be observed that agents within the same group quickly converge to similar behaviors when the game comes to the second round, as indicated by their reported numbers falling within a narrow range. Agents using Qwen2-72B and Llama3-70B exhibit relatively consistent behaviors, while some agents using MistralAI-8×22B might exhibit different behaviors, such as reporting larger numbers.

The system prompt used in the group-level simulations is shown in Appendix G, in which we also provide several examples of agents' behaviors for better understanding. These examples show how agents consider the reported numbers from other groups and strategically choose the reported numbers to benefit their own group. Such a phenomenon confirms that agents can perform reasonable thoughts and actions to help achieve a collective goal.

## 4.6 FURTHER DISCUSSIONS

We provide further discussions on usage tips and open questions when conducting large-scale agent-based simulations.

**Impact of the Prior Knowledge of LLMs**  As "guess the $\frac{2}{3}$ of the average" is a classic game, it is not surprising that LLMs might have acquired prior knowledge from their training corpus. To measure the impact of this prior knowledge, we change the ratio from $\frac{2}{3}$ to $\frac{1}{2}$ and $\frac{51}{100}$, and then conduct the simulation experiments, respectively. Note that changing the ratios does not alter the fundamental nature of this game, and as a result, the behaviors (such as the reasoning process) of the participating agents are expected to remain similar if LLMs indeed understand the game.

The experimental results are shown in Fig. 8, from which we can observe that there are significant differences in the agents' performance when the ratio is set to $\frac{1}{2}$ and $\frac{51}{100}$, although both cases should be very similar. We observe that more agents tend to report large numbers (*e.g.*, around 50) when the ratio is $\frac{51}{100}$ compared to the scenario with a ratio of $\frac{1}{2}$, which indicates that some agents might not follow the game when setting the ratio to $\frac{51}{100}$. In response, we add a note into system prompts to encourage LLMs to draw from the classic game, stating *This game is a variation of the famous "guess the 2/3 of the average" game*. The results summarized in Fig. 8, denoted as "$\frac{51}{100}$+note", show that the winning number decreases from $11.85$ to $6.46$, aligning more closely with that of using the ratio $\frac{1}{2}$, reporting $6.21$. These experiments highlight the impact of LLMs' prior knowledge, and the effects of using a prompt to explicitly guide the agents and help them understand the settings of simulations.

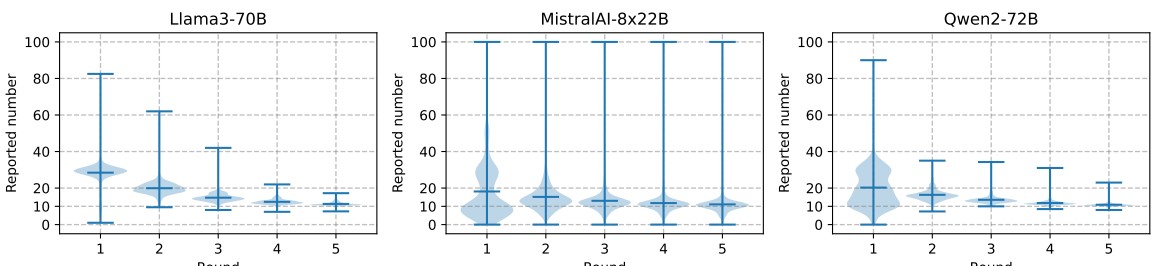

Figure 8: The distributions of reported numbers when setting different ratios in the game. We use "+note" to denote that we add a note to the system prompts.

Figure 9: The distributions of reported numbers when we change the Nash equilibrium to 10 in the simulation.

**When the Nash Equilibrium Is Not 0** We also set up another variant of the game to validate the agents' capabilities in understanding and reasoning. Specifically, we modify the winning criteria so that the Nash equilibrium becomes 10, instead of 0 as in the classic game. As a result, LLMs might not exhibit reasonable thoughts and behaviors if they have a limited understanding of this game or have poor reasoning ability. The adopted prompt is shown in Appendix H.1.

As shown in Fig. 9, we observe that some agents using the Qwen2-72B and MistralAI-8×22B are able to reason out the Nash equilibrium point in the first round. As the game progresses, the numbers reported by these agents gradually approach 10, demonstrating their understanding of this game and ability to make reasonable decisions.

The above findings are further confirmed by observations of individual-level behaviors shown in Appendix H.1. Besides, from these observations, we can identify some typical mistakes made by agents. For example, some agents might make simple calculation errors, such as calculating a wrong average value, and leading to incorrect results despite having a correct reasoning process. Some agents may follow a logical step-by-step process (*e.g.*, calculating the Nash equilibrium) but still make incorrect decisions (*e.g.*, directly reporting 0). But even so, these mistakes are infrequent and do not affect the overall conclusions drawn from group-level observations, as we previously demonstrated.

These observations indicate that although agents may make simple calculation errors (especially in decimal calculations), they exhibit powerful reasoning abilities and show great potential for usage in large-scale simulations. More experimental results and further discussions regarding the impact of temperature and the role-playing ability of LLMs can be found in Appendix H.2 and H.3, respectively.

## 5 CONCLUSIONS

In this paper, we first discuss several key factors of concern for conducting large-scale agent-based simulations, including scalability and efficiency, population distribution and agent diversity, and ease of management. Motivated by these factors, we propose and implement several enhancements based on AgentScope, including an actor-based distributed mechanism that provides agent-level parallel execution and automatic workflow conversion, the flexible environment support to simulate various real-world scenarios, the heterogeneous configurations that allow users to specify population distributions and to automatically generate agents with diverse background settings, and a web-based interface to simplify the management of large-scale agents. These enhancements make the framework more flexible and convenient for supporting large-scale agent-based simulations. We conduct a series of simulation experiments with the proposed framework and provide detailed observations on the diverse and realistic behaviors of agents, highlighting its great potential to further advance research and applications in agent-based simulations. We will release the source code to inspire further research and development in large-scale multi-agent simulations.

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

## A    ETHICS STATEMENT

This study is concentrated on proposing a novel multi-agent framework designed to assist researchers in conducting various simulation experiments, drawing valuable behavioral observations and insights to promote the development of related fields. We are committed to upholding principles of equity and fairness, and firmly reject any form of prejudicial discrimination based on age, education level, occupation, or any other characteristics.

## B    EXAMPLES

### B.1    EXAMPLE OF AGENT-LEVEL PARALLEL EXECUTION

An example of agent-level parallel execution is shown in Fig. 10, where agent-B and agent-C both rely on the messages from agent-A, allowing them to be executed in parallel once the execution of agent-A is completed. In contrast, agent-C and agent-D cannot be executed in parallel, as agent-D depends on messages from agent-C.

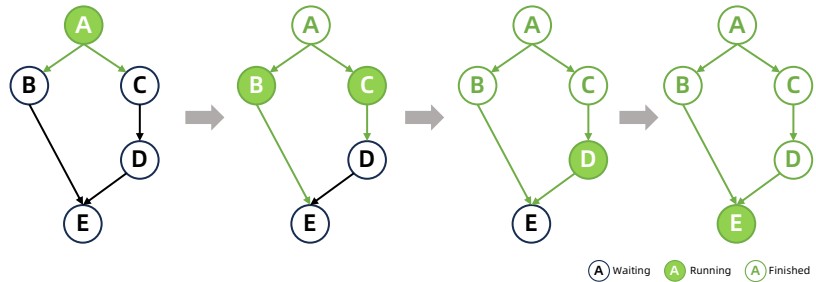

Figure 10: An example of agent-level parallel execution, where circles represent agents and directed edges represent message passing flows.

### B.2    EXAMPLE OF MULTIPLE ENVIRONMENTS

An example of a multi-layer environment structure designed for group-wise information synchronization is shown in Fig. 11. Different environments can be established for different groups of agents to provide interactive items and shared information. Upon these environments, a global environment can be configured for global synchronization. Such a multi-layer environment structure can be employed in simulations that necessitate both intra-group collaborations and inter-group information differentiation.

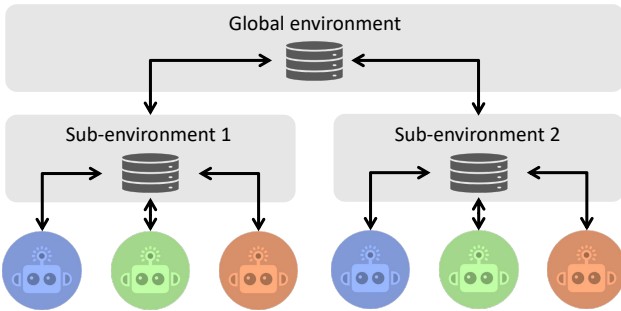

Figure 11: A multi-layer environment structure for agent-based simulation.

### B.3    EXAMPLE OF CONFIGURATION FILE

In Listing 1, we show an example of the configuration file for a group of people with different educational levels.

```
1   # The high level parameters
2   population: 1000
3   # distribution configuration
4   distributions:
5     - name: "Education Level"
6       categories:
7         - name: "Elementary School"
8           proportion: 0.2
9         - name: "High School"
10          proportion: 0.2
11        - name: "Bachelor's Degree"
12          proportion: 0.2
13        - name: "Master's Degree"
14          proportion: 0.2
15        - name: "Ph.D."
16          proportion: 0.2
17    - name: "Gender"
18      categories:
19        - name: "Male"
20          proportion: 0.5
21        - name: "Female"
22          proportion: 0.5
23      # ...
```

Listing 1: Example of configuration file for people with different educational level.

### B.4 EXAMPLE OF THE WEB-BASED VISUAL INTERFACE OF AGENT-MANAGER

As illustrated in Fig. 12, a web-based visual interface provides a comprehensive overview of all registered servers and all deployed agents on different devices, from which users can view the server's identity, IP address, running status, and utilization of computing resources.

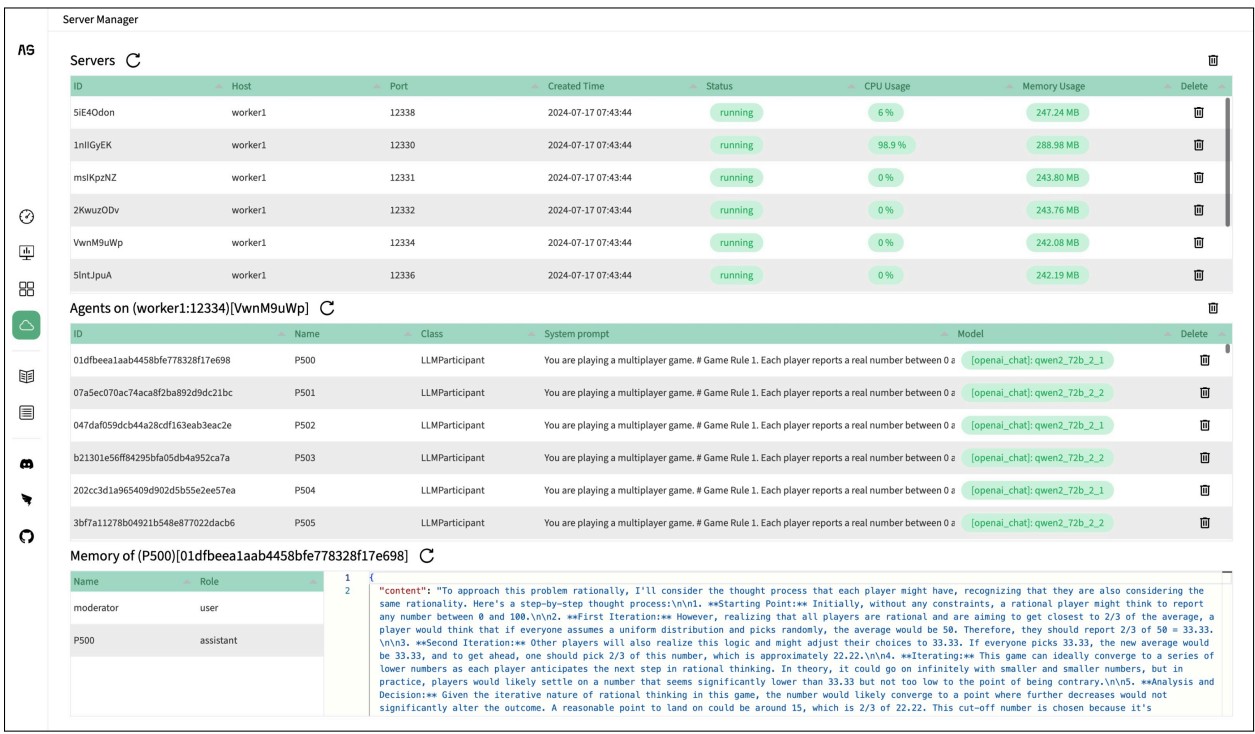

Figure 12: The web-based visual interface for large-scale agents management.

## C Experimental Settings

### C.1 Details of Adopted LLMs

The details of the adopted LLMs are provided below:

- **Llama3-8B / Llama3-70B** (Meta, 2024): A series of open-source LLMs developed by Meta, which have been pre-trained and fine-tuned on a massive corpus.

- **Qwen2-7B / Qwen2-72B** (Yang et al., 2024a): The second generation of Qwen open-source LLMs, developed by Alibaba.

- **MistralAI-8x7B / MistralAI-8x22B** (MistralAI, 2024): The open-source mixture-of-experts (MOE) LLMs released by MistralAI, where each MOE LLM consists of eight 7B/22B models.

### C.2 Implementation Details

Due to the limited GPU memory, unless otherwise specified, we deploy eight Qwen2-7B / Llama3-8B models, two Qwen2-72B / Llama3-70B / MistralAI-8x7B models, or one MistralAI-8x22B model on each device. The generation temperature for all LLMs is set to $1.0$ to promote the diversity of responses. Besides, to prevent errors in response format, each agent executes two LLM server calls in every game round. The first call is used to generate the thought process and the reported number, while the second call is made to extract the reported number correctly.

## D The Number of Response Tokens

The statistics of the response tokens of agents when using different LLMs and system prompts can be found in Fig. 13.

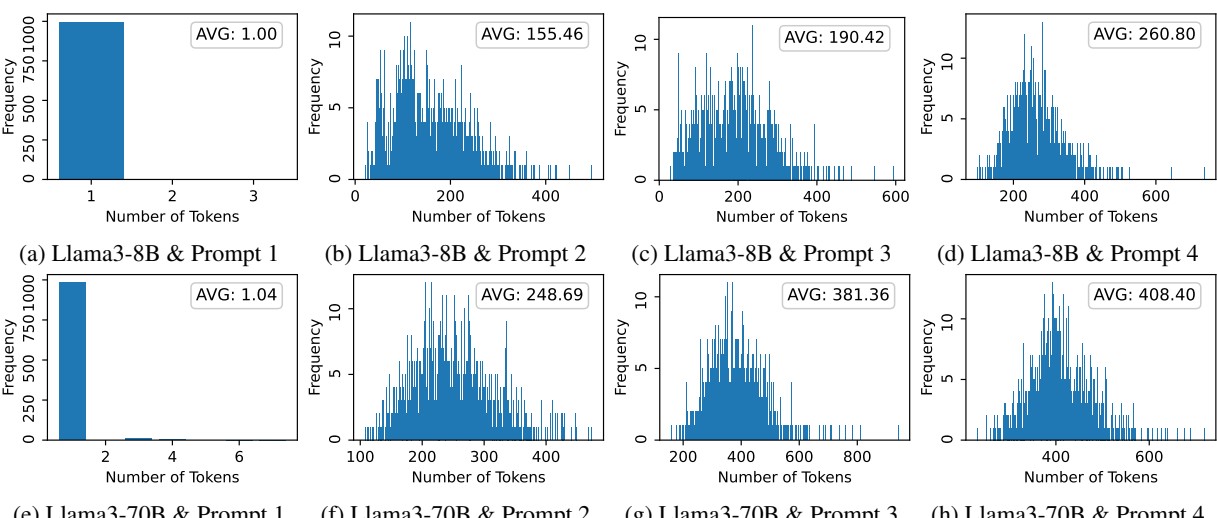

(a) Llama3-8B & Prompt 1    (b) Llama3-8B & Prompt 2    (c) Llama3-8B & Prompt 3    (d) Llama3-8B & Prompt 4

(e) Llama3-70B & Prompt 1    (f) Llama3-70B & Prompt 2    (g) Llama3-70B & Prompt 3    (h) Llama3-70B & Prompt 4

Figure 13: The response tokens when using different LLMs and system prompts.

## E Responses of Agents with Different LLMs and System Prompts

### E.1 Distributions and Statistics of the Reported Numbers

The distributions of the reported numbers of agents with different LLMs and system prompts are shown in Fig. 14, which includes the average (AVG), minimum (MIN), maximum (MAX), standard deviation (STD), median (MED), and mode (MOD). Besides, in Fig. 15, Fig. 16, and Fig. 17, we report the distributions of the reported numbers of agents with different LLMs and prompts in a multi-round game.

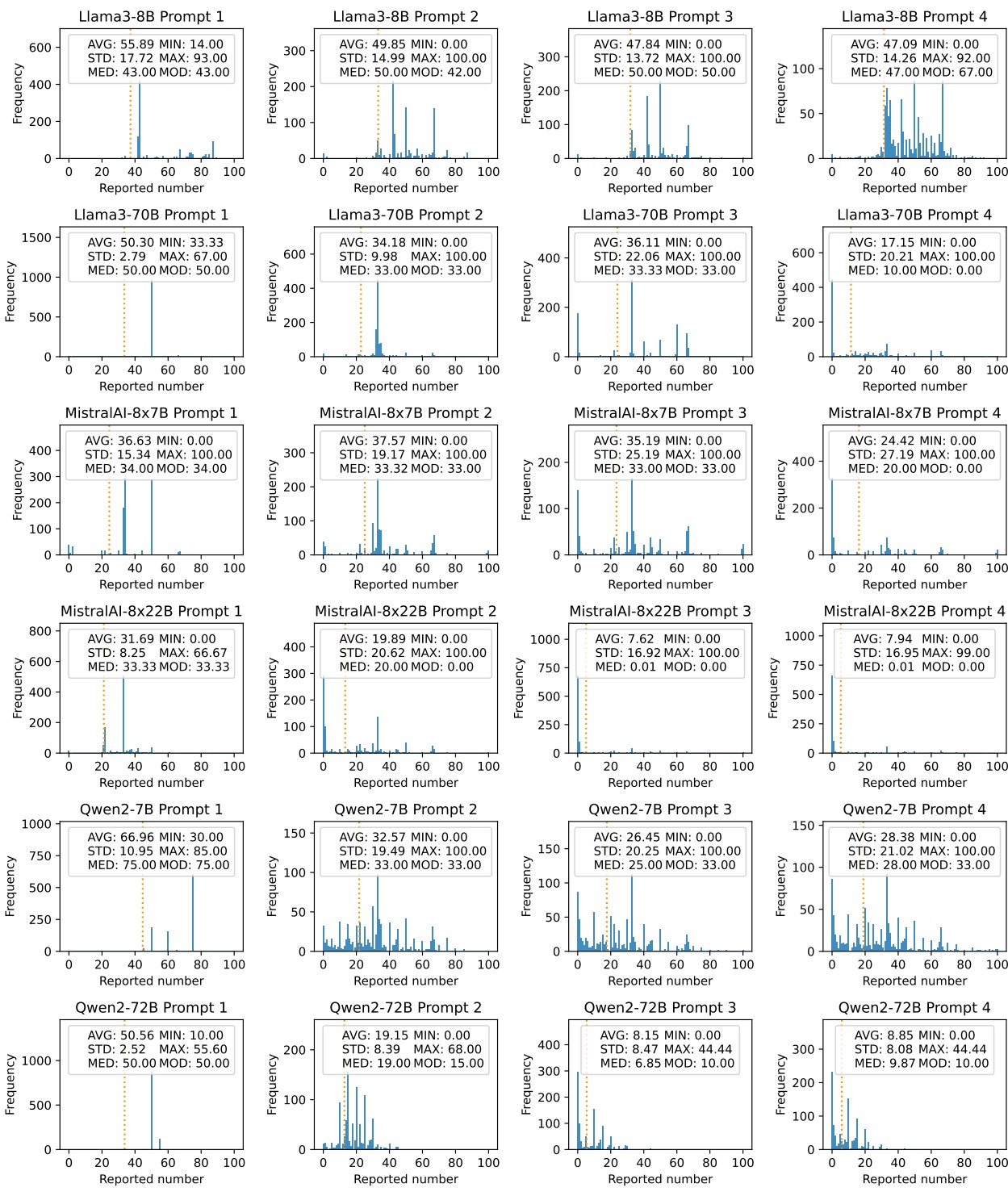

Figure 14: The distributions of the reported numbers of agents with different LLMs and prompts. The horizontal axis represents the reported numbers, and the vertical axis represents the frequency of occurrence of each number. The blue bars represent the distributions of reported numbers, while an orange dashed line indicates the winning number.

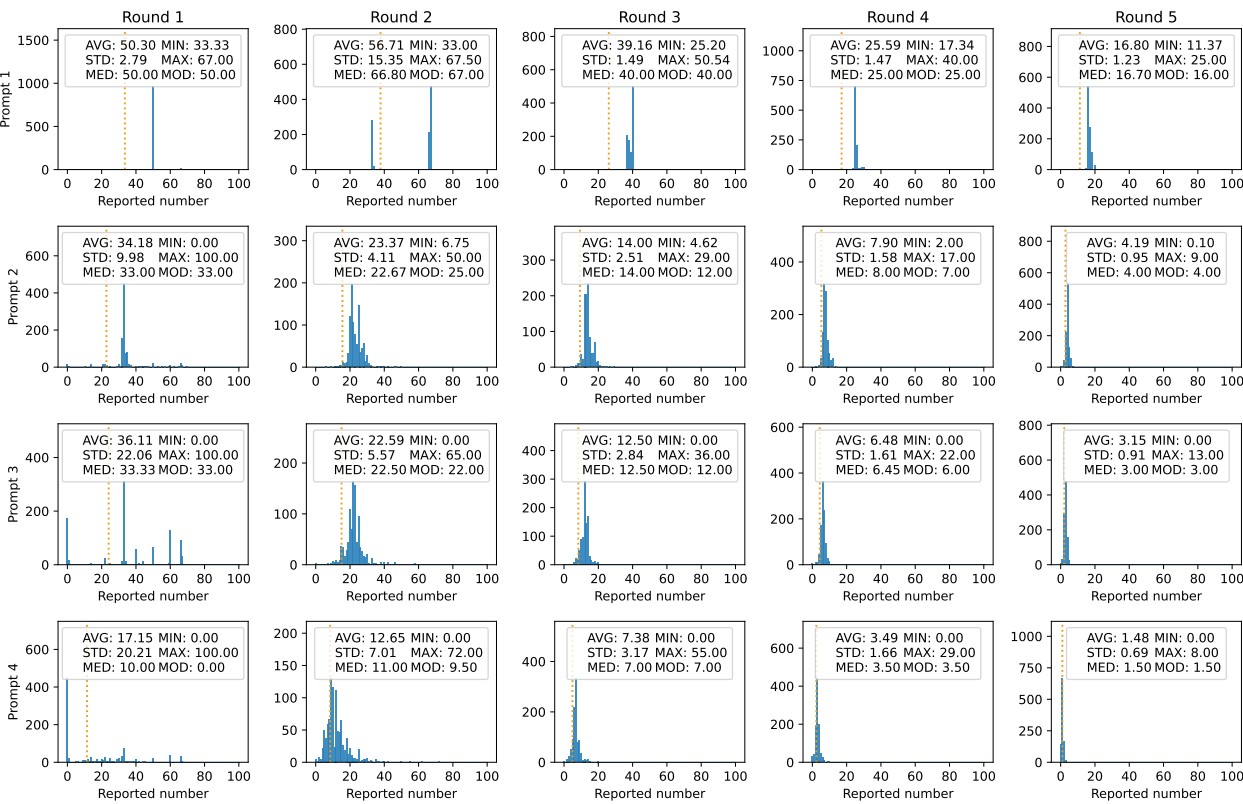

Figure 15: The distributions of the reported numbers of agents with Llama3-70B in a multi-round game.

## E.2 INDIVIDUAL-LEVEL OBSERVATIONS AND CASE STUDIES

For better understanding, we provide examples of the responses of different LLMs with Prompt 1 and Prompt 2, as shown in Example 1 and Example 2, respectively.

---

**Example 1: Responses of agents employing different LLMs with Prompt 1**

**Llama3-8B**: 43
**Llama3-70B**: 50
**Qwen2-7B**: 75
**Qwen2-72B**: 50
**MistralAI-8x7B**: 34
(This number is based on strategic estimation and the desire to win the game, considering the given rule.)
**MistralAI-8x22B**: Based on the rules you've provided, my strategy would be to predict what numbers other players might report and then calculate the average accordingly. However, since I can't interact with other players or know their strategies, I'll make an informed guess. If all players report a number close to 50, the average would be around 50, and 2/3 of that would be approximately 33.33. So, I would report 34 to maximize my chances of being the closest.

---

In a multi-round game, we observe that agents can consider the winner number from the previous round when reporting numbers in the current round. For example, as shown in Example 3, some agents tend to report a number slightly smaller than the winner number, while others choose to report $\frac{2}{3}$ of the winning number from the previous round, as illustrated in Example 4.

## E.3 DETAILED INSTRUCTIONS IN SYSTEM PROMPTS

To further explore the impact of behavioral guidance on agents, we incorporate more detailed instructions tailored for this game in the system prompts. Specifically, we remind agents that all their competitors are rational and will try to adjust the reported numbers by analyzing others' strategies, resulting in Prompt 3 and Prompt 4 respectively. With

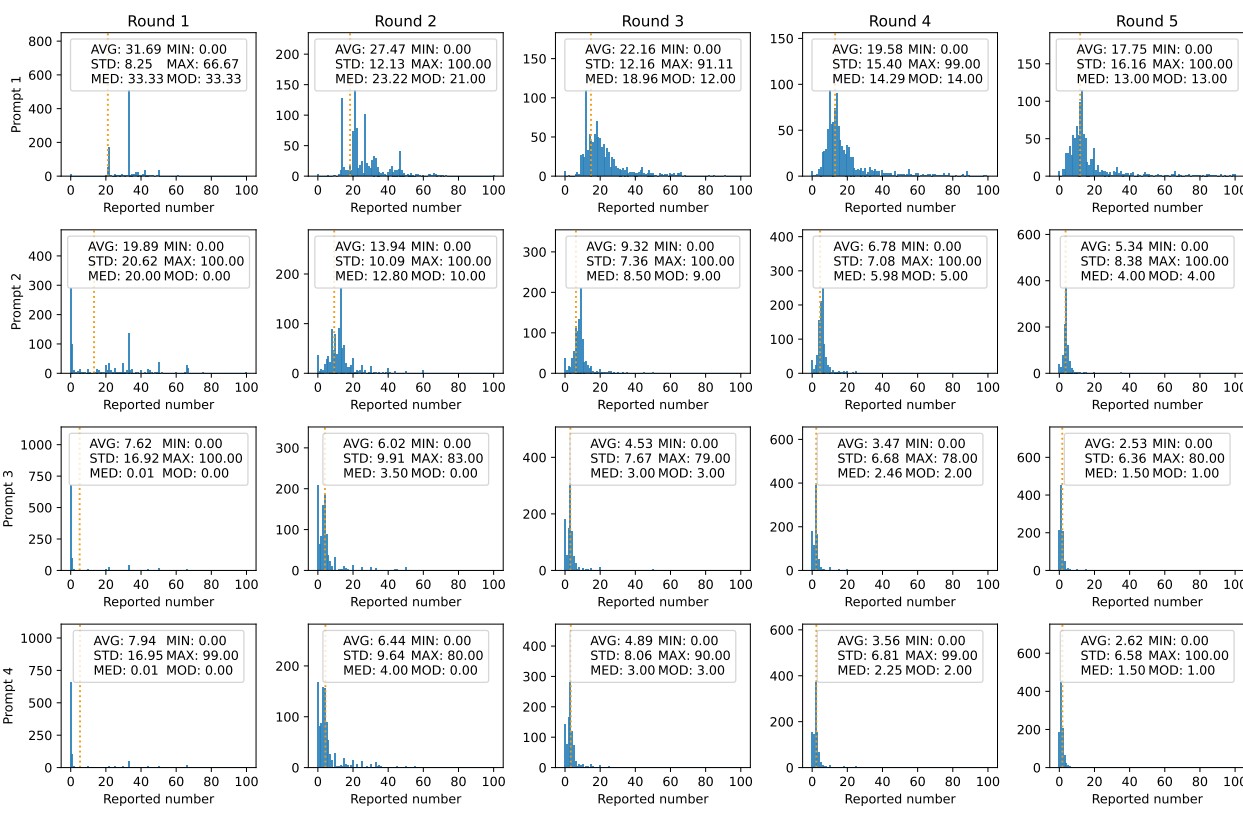

Figure 16: The distributions of the reported numbers of agents with MistralAI-8×22B in a multi-round game.

adding such behavioral guidance in the system prompts, we expect agents can engage in more thoughtful and diverse considerations before reporting their numbers, thereby making simulations more practical, meaningful, and interesting.

---

**Prompt 3**

You are playing a multiplayer game.

# Game Rule
1. Each player reports a real number between 0 and 100, inclusive.
2. The winner will be the player whose number is the closest to 2/3 of the average of all reported numbers.

# Note:
1. All players are rational.

Think step by step and then report your number.

---

**Prompt 4**

You are playing a multiplayer game.

# Game Rule
1. Each player reports a real number between 0 and 100, inclusive.
2. The winner will be the player whose number is the closest to 2/3 of the average of all reported numbers.

# Note:
1. All players are rational.
2. All players will try to guess the others' strategies to adjust their own strategies.

Think step by step and then report your number.

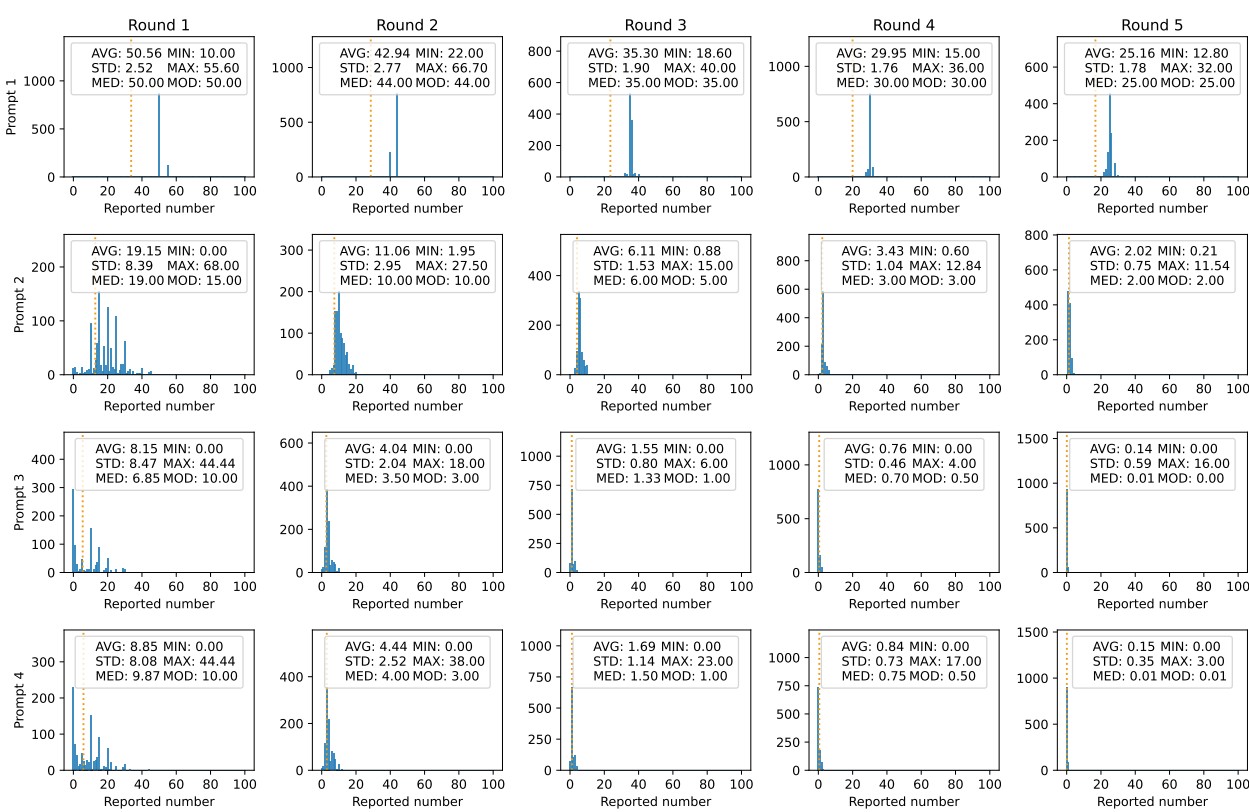

Figure 17: The distributions of the reported numbers of agents with Qwen2-72B in a multi-round game.

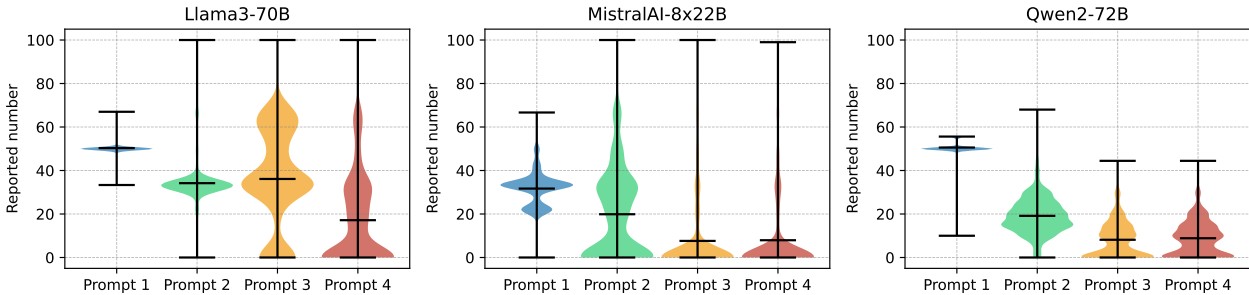

Figure 18: The impact of different system prompts on different LLMs.

The comparisons among different system prompts are illustrated in Fig. 18. In general, from the figure we can observe that the reported numbers are closer to 0 when using Prompt 3 and Prompt 4 than those of Prompt 1 and Prompt 2. These experimental results indicate that detailed instructions are more effective than general guidance (*e.g.*, "think step by step") in encouraging agents to perform thoughtful considerations and take rational actions. Several case studies shown in Example 5 and Example 6 confirm the effectiveness of providing detailed instructions in system prompts. These instructions encourage agents to perform reasonable behaviors by guiding them to regard other agents as rational players (as instructed by Prompt 3) and to guess other agents' strategies (as instructed by Prompt 4).

Furthermore, in a multi-round game illustrated in Fig. 19, agents using Prompt 3 and Prompt 4 can converge to the Nash equilibrium faster than those using Prompt 1 and Prompt 2. For example, agents with Qwen2-72B report 35.30, 6.11, 1.55, and 1.69 in average at the third round when using Prompt 1, Prompt 2, Prompt 3, and Prompt 4, respectively, while in the fifth round, the average of reported numbers become 25.16, 2.02, 0.14, and 0.15.

It is worth noting that the impact of the system prompts on different LLMs can be different. For example, from the perspective of the range of the reported numbers (*i.e.*, the maximum and minimum value of reported numbers among all agents), employing Prompt 3 and Prompt 4 in Qwen-72B can significantly reduce the maximum number, while that of

---

**Example 2: Responses of agents employing different LLMs with Prompt 2**

**MistralAI-8x22B:** This is a classic game theory problem known as the "Two-Thirds of the Average Game". The Nash equilibrium strategy for this game is to choose 0.
Let me explain step by step:
1. Suppose the other players follow the strategy of choosing 2/3 of the average of all reported numbers.
2. If you choose any number larger than 0, for example, suppose you choose 'x' (0 < x <= 100).
3. Because the other players choose 2/3 of the average, the average of all reported numbers will be slightly larger than 2/3 of the average. Therefore, the true value the player should be aiming for is not 2/3 of the average, but slightly lower.
4. This indicates that the number 'x' cannot be the optimal strategy for the player, because the true value the player should aim for is slightly lower than 2/3 of the average.
5. Hence, the optimal strategy for a player is to choose 0, since this is the only number that will make the player's number equal to 2/3 of the average.
So, I will report my number as 0.

**Llama3-70B:** Let's think this through step by step.
*Step 1: Understand the game**
The game asks each player to report a real number between 0 and 100, inclusive. The winner is the player whose number is closest to 2/3 of the average of all reported numbers.
*Step 2: Analyze the game structure**
The game has a similar structure to the "Guess 2/3 of the average" game, a classic example of a game that exhibits a Nash equilibrium. In this type of game, the optimal strategy is to guess the 2/3 of the average of the expected numbers.
*Step 3: Estimate the expected numbers**
Since each player can report any number between 0 and 100, a reasonable estimate of the expected numbers is the midpoint of the range, which is 50.
*Step 4: Calculate the average of the expected numbers**
Assuming all players report numbers around 50, the average of the reported numbers will be close to 50.
*Step 5: Calculate 2/3 of the average**
2/3 of 50 is approximately 33.33.
*Step 6: Report the optimal number**
Based on the analysis, I will report the number that is closest to 33.33, which is **33**.
So, my reported number is **33**.

**Qwen2-72B:** To think step by step through the logic of this game, we need to consider how other players might think as well. This problem is a classic example of a game theory scenario known as the "2/3 of average game" or "Guess 2/3 of the Average."
1. **First Level Thinking**: If players think purely randomly or without regard for strategy, they might pick any number between 0 and 100. Let's assume the average number picked by such players would be 50.
2. **Second Level Thinking**: If a player assumes most players are thinking at the first level and picks 50, then the smart strategy is to pick 2/3 of 50, which is 33.33.
3. **Third Level Thinking**: If players think the majority will think of 33.33, then choosing 2/3 of 33.33, which is roughly 22.22, might win.
4. **Fourth Level Thinking**: Continuing this argument, players might anticipate others to pick 22.22 and choose 2/3 of 22.22, which is about 14.81.
This process could theoretically continue on and on, with each level of thinking reducing the anticipated value to pick by around 33% (2/3).
However, in a real game scenario, we can't expect everyone to think to the same level of depth, and the complexity of these calculations can quickly provide diminishing returns. Thus, in practice, the iteration shown above is typically where a theoretical convergence starts to happen.
Given these considerations, a reasonable choice to make in this game, assuming many players think similarly to this level and you want to outthink them, could be to pick around 15. However, to give a more specific number, let's round to avoid giving exact theoretical values that others might also settle on.
*I will report: 14.**
This number assumes a level of collective higher-order thinking among players and aims for a value just under common round numbers like 15, in an effort to be slightly unpredictable while still applying strategic thought.

---

Mistral-8×22B remains unchanged. Besides, using detailed instructions in system prompts might increase the token number of responses, as summarized in Appendix D, because agents are likely to consider multiple aspects before providing an answer.

---

**Example 3: Agent behavior: reporting a number slightly smaller than the winning number in the previous round**

Given the outcome of the previous round, it's clear that the winning number was close to the prediction made in the initial round of thinking through several levels of 2/3 averages. The winner number of 12.77 suggests that other players also employed a multi-layered strategy of rational thinking, pushing the 2/3 of the average down to a lower range.
### Strategy Review:
- In the previous round, I chose 13 based on the assumption that other players would also engage in several levels of recursion, anticipating that the selection would converge towards a number in the low teens.
- The actual winning number was 12.77, aligning closely with this strategy.

### Adjusting Strategy for the Next Round:

- The game dynamics suggest that the players may continue to employ similar levels of strategic thinking, aiming for numbers around the same range.
- To refine the strategy:
- We must acknowledge that the group's rational level seems quite advanced, given the convergence towards the low teens.
- We also assume that players will notice the pattern and may adjust their next numbers based on this outcome.

### Predicting the Next Outcome:

- If all players think that the last round's average had a significant impact, there might be a slight adjustment towards higher or lower numbers based on feedback learning.
- Assuming players understand and trust the pattern, they may converge slightly higher or stay very close to the previous level, expecting a similar game state.

### Final Decision:

Given the feedback from the most recent game state, I will aim for a number close to 12.77, but with a slight adjustment to avoid direct match and account for any small behavioral shifts in other players.

**New Reported Number for the Next Round: 12.7**

This number aims to stay within the revealed strategic range, while making a conservative adjustment based on the observational feedback from the last round. It attempts to balance the possibility of convergence and the need to differentiate from the most recent winning number, which may influence players' decisions in this round.

---

**Example 4: Agent behavior: reporting $\frac{2}{3}$ of the winning number from the previous round**

Given the winner number from the last round was 12.77, we can infer a few points:
1. When 2/3 of the average led to 12.77, it implies the aggregate average must have been slightly under 19.155 (since 12.77 is approximately 2/3 of 19.155).
2. It seems that players chose numbers that led to this average, suggesting a collective reasoning pattern typical in the Centipede Game or related games—players iteratively reduce their number based on the assumption that others will also reduce their numbers based on the logic above.
Continuing from 14.8 in the last round, taking into account this feedback and assuming players will further lower their expectations after seeing the average result, one might consider reducing the guess by another iteration. I'll simulate one further level of reduction to find my next guess:

- If players previously aimed for around 19.155 (which led to the average and then the winning guess of 12.77 when adjusted by 2/3), applying the same logic would first estimate the next middle ground by assuming players will adjust down to align with 12.77 as the new reference point.
- Taking 2/3 of 12.77 yields approximately 8.51 (2/3 * 12.77 ≈ 8.51).

Given this potentially iterative mechanism, and thus expecting other players might also think in this direction, I will use 8.51 as the new central guess point, informed by the winner number of the previous round and the hypothesis that the collective reasoning would continue on a similar path.

My reported number is 8.51.

---

**Example 5: Agent behavior: regarding agents as rational players**

This game is a classic example of a reverse induction game. Here's the logic behind the rational choice:

1. If all players think completely rationally and choose numbers uniformly at random, the average would be 50.

2. However, knowing that other rational players can also think this way, one would realize that if everyone picks 50, the target to aim for is 2/3 of 50, which is approximately 33.33.

3. But if everyone then adjusts their number to around 33.33, the rational choice would then be 2/3 of 33.33, which is about 22.22.

4. Continuing this line of reason, this iterative process drives the optimal choice towards zero. However, the problem nuances that it can't go below zero since it's bound between 0 and 100.

Considering this, and if we assume the game has been repeated indefinitely in the players' imaginations, the logical steady-state answer would be 0, as it is the limit point of the iterative 2/3 adjustment process from the starting point of 50.

Given that I am playing this game and assuming all participants are rational and have reached a similar conclusion, reporting 0 seems to be a strategic decision.

**Therefore, I report my number as 0.**

---

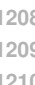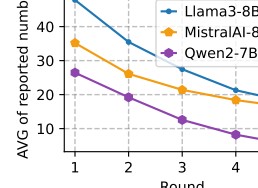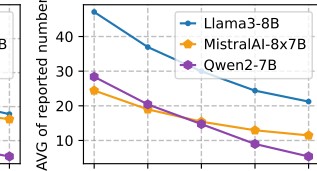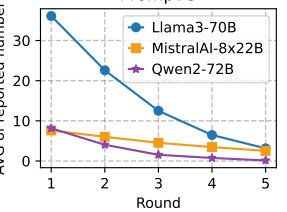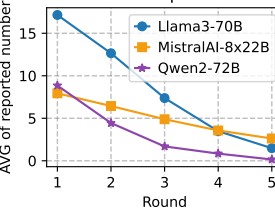

Figure 19: The average of reported numbers in the multi-round games.

# F    ADDITIONAL RESULTS FOR AGENTS WITH DIVERSE BACKGROUNDS

## F.1    PROMPT FOR GENERATING BACKGROUND AND EXAMPLES

The meta prompt used for generating heterogeneous background settings can be found in Prompt 5.

---

**Prompt 5**

You need to generate a person's background description based on the user-provided JSON format information.
In addition to the information provided by the user, each background description must also include the person's name, age, gender, job, and a paragraph describing the character's personality.
Please output the background description after "## Background" tag.

$\{JSON\}$

---

We use GPT-4o to generate these diverse yet detailed background settings with Prompt 6. We provide two generated background settings for agents, including a character who has a bachelor's degree ( Example 7) and a character who is an artist ( Example 10).

---

**Example 6: Agent behavior: guessing other agents' strategies**

To solve this problem, let's consider the thought process of each rational player involved. Since all players are rational, they will assume that everyone will follow the same line of reasoning.

1st Level Thinking:
At the first level of thinking, a player might assume that everyone will choose numbers at random between 0 and 100. Therefore, the average of all reported numbers would be approximately 50. Following the rule, choosing 2/3 of 50 (or 33.33) would be optimal.

2nd Level Thinking:
Now, recognizing that all players are rational, one might assume that everyone will think about the first level of thought and hence choose 33.33. So, the second level of thinking would suggest to choose 2/3 of 33.33 (or approximately 22.22).

3rd Level Thinking:
In the same vein, if everyone is thinking on the second level and will choose about 22.22, then 2/3 of that would lead you to choose about 14.81.

This process of iteratively calculating 2/3 of the previous average can be continued to reach a number that all rational players might converge on. This iterative process approaches a limit, which can be determined mathematically.

Let's denote the number we're trying to find as X. The average initially is 50, which leads us:
X = (2/3) * 50

Each subsequent iteration uses the previous iteration's result:
X = (2/3) * X

Solving for X in the equation X = (2/3) * X leads us to the result that X approaches 0 as the number of iterations goes to infinity. However, there's a minimum number that can be chosen given the problem setup, so the optimal number a rational player might choose, under the assumption that everyone is thinking similarly and converging to this limit, would be close to zero, but the game rules indicate that the number is between 0 and 100 and it is rational to assume that not all players will reach this last step of thought.

Given the recursion and the rational view of all players involved, a number very close to zero is the most rational choice a player can make, albeit with a minimum margin to allow for slight deviations from pure logic in reality and as to not necessarily report 0 exactly, which might be chosen by other players leading to a tie or unpredicted result.

Let's say I'll choose a number slightly above zero, such as 0.1, assuming all players are going through a complex but similar thought process. This choice reflects a belief that other players will choose similarly rational but not necessarily identical numbers, resulting in an average close to zero but not exactly zero.

Therefore, my reported number will be 0.1.

---

**Prompt 6**

You are playing a role in a multiplayer game, make sure your behavior fits the following character background.

# Character Background

{*background*}

# Game Rule
1. Each player reports a real number between 0 and 100, inclusive.
2. The winner will be the player whose number is the closest to 2/3 of the average of all reported numbers.

# Note
1. Please strictly follow your character background in the game.

Think step by step and then report your number.

---

## F.2 CASE STUDIES ON AGENTS CHARACTERIZED WITH DIFFERENT BACKGROUND SETTINGS

We provide case studies on individual-level behaviors when providing detailed yet diverse background settings. For example, we show the responses of agents characterized as an elementary school student in Example 8 and a character

---

**Example 7: A generated background setting profiling a character who has a bachelor's degree**

Name: Thomas Reed
Age: 29
Gender: Male
Job: Software Engineer
Education Level: Bachelor's Degree

Thomas Reed is a meticulous and driven individual with a Bachelor's Degree in Computer Science. He has a sharp analytical mind and an insatiable curiosity for technology and coding. Known for his problem-solving skills and creativity, he thrives in environments that challenge his intellect and foster innovation. Thomas is also a dedicated team player who values collaboration and often goes out of his way to help his colleagues. Outside of work, he enjoys learning about the latest tech trends and contributing to open-source projects.

---

who gets the Ph.D. degree in Example 9. From these cases, we observe that agents can effectively perform reasoning processes and make decisions according to their assigned roles.

---

**Example 8: Agent behavior: playing as an elementary school student**

Considering my character is Emily, a 10-year-old student, her thought process would be simple, curious, and somewhat less complex than an older player's. Here's how she might think through the game:

1. I need to pick a number between 0 and 100.
2. The game says to guess what 2/3 of the average of all numbers will be. 3. If everyone picks a number around 50 because it's in the middle, then the average will be about 50.
4. Two-thirds of 50 is... um... that's a little tricky. Let's see, 1/3 of 50 is about 16 (I think it's 16 and 2/3, but that doesn't really matter), so 2/3 would be about double that, right? So, it's... 32!
5. But if everyone thinks like that, then maybe I should pick a little less than 32 because they might guess a little less.
6. I think I'll choose 30 because it's close to the number I calculated and it sounds nice.

Emily decides to report the number **30** based on her simple calculation and intuition.

---

**Example 9: Agent behavior: playing as a Ph.D.**

Considering the game rule, as a research scientist, I would approach this logically and strategically. Knowing the target is to be closest to 2/3 of the average of all numbers reported, it's likely the average would be somewhere around 50 if everyone picks a number randomly. To be precise and consider the scientific reasoning, I would think step-by-step:

1. The lowest possible number is 0 and the highest possible number is 100.
2. If everyone thinks rationally, they would not pick 0 or 100 because it's highly unlikely that everyone would pick the same number.
3. The first average guess might be around 50, which always acts as a starting point in such contexts.
4. Since I need to be closest to 2/3 of the average, starting with the idea that others might think rationally, the 2/3 of 50 would be about 33.3.
5. However, if everyone thinks I think this way, then the average is likely to be much lower, around 33.3, and I should aim for 2/3 of 33.3, which is about 22.2.
6. Continuing this iterative logical process, it's a game of outthinking the common logic at each step.

As someone who works extensively with statistics and probabilities in my research, I would apply these principles. My final number, considering the likely average guess and the repeated application of the 2/3 rule based on the rationality of others, would be:

15.

This choice reflects a scientific and analytical approach, attempting to predict and counter the expected behavior of other players.

---

### F.3 AGENTS WITH DIFFERENT OCCUPATIONS

We also conduct a simulation experiment involving agents with different occupations, in which we assign six different occupations to agents, including professors in game theory, economists, psychologists, athletes, artists, and writers. An example of the generated background setting of an artist can be found in Example 10.

> **Example 10: A generated background setting profiling a character who is an artist**
>
> Name: Thomas
> Age: 34
> Gender: Male
> Job: Artist
> Thomas is a dedicated artist with a unique vision and a passion for creating thought-provoking works. At the age of 34, he has already built a reputation for his innovative approach to contemporary art. Thomas's personality is characterized by his profound sensitivity and introspective nature, which is evident in each piece he creates. He has an exceptional ability to observe the world around him, finding beauty in places others might overlook. This keen sense of observation is paired with a strong sense of empathy, allowing him to connect deeply with his audience. Thomas is also known for his humility and modesty, often shying away from the spotlight and letting his art speak for itself.

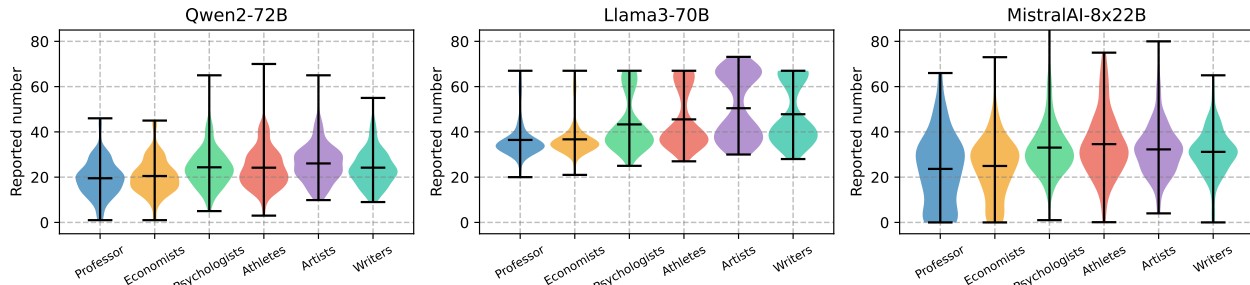

Figure 20: The distributions of numbers reported by agents characterized with different occupations.

As demonstrated in Fig. 20, the experimental results confirm the impact of the different occupational descriptions assigned to the agents. It can be concluded that agents characterized as professors in game theory and economists tend to report smaller numbers than other agents. Some case studies on individual-level behaviors, as shown in Example 11, further confirm that agents' considerations and actions are consistent with their respective occupations. For example, agents characterized as professors in game theory tend to adjust their behaviors under the assumption that others may not be rational enough, while those characterized as artists might adopt a straightforward strategy.

> **Example 11: Agent behavior: playing as a professor in game theory**
>
> Given the game situation, I will approach this as a puzzle requiring strategic thought, which aligns with my expertise in game theory. First, let's assume that there are no constraints on the strategic reasoning of the participants, meaning they can all engage in rational decision-making.
>
> If all players were to report a number between 0 and 100 without any strategic consideration, we could theoretically have an average of 50 (midpoint of the range). However, knowing that the goal is to guess 2/3 of the average, the first level of strategic reasoning would be to guess 2/3 of 50, which is 33.33.
>
> But if all players are rational and think similarly, everyone would guess 33.33, leading to the actual target number being 2/3 of 33.33, which is approximately 22.22. This pattern of reasoning continues, with each level of rationality leading to a lower guess, as players recursively apply the 2/3 rule to the previous average guess.
>
> Considering that I am a professor of game theory and likely have a higher level of strategic thinking compared to other players, I will assume that many players will arrive at the second level of reasoning (guessing 22.22) or possibly even higher. However, to avoid overthinking and potentially overshooting the expected target number, I will stick with a prediction at the third level of recursion, which is to guess 2/3 of 22.22, leading to a guess of approximately 14.81.
>
> Thus, I will report the number **15**, as it is a rounded and practical choice that should be close to the 2/3 of the average if most players engage in at least two levels of strategic consideration.

# G    ADDITIONAL RESULTS FOR MIXTURE OF LLMS

## G.1    INDIVIDUAL-LEVEL SIMULATION

Example 12 shows a typical response in the individual-level simulation involving a mixture of LLMs, where an agent report chooses a number slightly smaller than the winner number in the previous round.

---

**Example 12: Agent behavior: reporting a number slightly smaller than the winner number in the previous round**

Interesting twist! It seems not all players followed the equilibrium strategy. However, let's think through how the previous result might affect our decision.

Since the winning number last round was 15.90, and this number is the closest to 2/3 of the average of the numbers reported by players, we can estimate the average of last round as:

15.90 * (3/2) = 23.85

Now assuming that players keep their previous strategies and report numbers close to the one they reported before, the average could still be around 23.85.

To maximize our chance of winning, we would report a number close to 2/3 of this estimated average:

2/3 * 23.85 ≈ 15.90

Since the game does not allow reports of the exact winning number from the previous round, let's choose a number close to this value, but not exactly equal to 15.90.

Let's report the number 15.89 for this round.

---

## G.2    GROUP-LEVEL SIMULATION

**System Prompt**    The system prompt used in group-level simulation when we adopt the mixture of LLMs setting (see Sec. 4.5) is shown in Prompt 7.

---

**Prompt 7**

You are playing a multiplayer game.

# Game Rule
1. There are 3 groups of players in the game.
2. Each player reports a real number between 0 and 100, inclusive.
3. Each group reports the average of all players in the group.
4. The winner will be the group whose number is the closest to 2/3 of the average of all groups' numbers.
5. You are in group $\{id\}$.

The 2/3 of the average for this round is $winner$. The numbers reported by groups are Group 1: $v_1$, Group 2: $v_2$, Group 3: $v_3$.
Let's move on to the next round.
Think step by step and then report your number.

---

**Examples of Agent Behaviors**    We provide three examples in Example 13, Example 14, and Example 15 to show the behaviors of agents in different groups. We observe that agents can perform reasoning processes considering the behaviors of other groups and how to contribute to their own group.

# H    ADDITIONAL RESULTS FOR FURTHER DISCUSSIONS

## H.1    WHEN THE NASH EQUILIBRIUM IS NOT 0

**System Prompt**    When we perform the variant game that the Nash equilibrium is not zero (see Sec. 4.6), the adopted prompt is shown in Prompt 8.

---

**Example 13: Behaviors of an agent in group 1 (MistralAI 8×22B)**

As we are moving to the next round, we should analyze the previous round's results to adjust our strategy. In the previous round, the winning condition was 2/3 of the overall average, which was 22.50. The groups reported the following averages:

Group 1: 39.36
Group 2: 21.14
Group 3: 40.75

Though no group perfectly met the winning condition, Group 2 had the closest average to the target of 22.50, making them the winner of the previous round. However, since we are aiming for the most optimal strategy, it's reasonable to assume that the other groups will also adjust their strategies for the next round. Based on the previous round's results, we can deduce that the overall average is currently around 33 (considering the winning condition of 22.50 as 2/3 of the overall average).

To increase the chances of your group winning the next round, you should aim to report a number closer to the winning condition, which is 2/3 of the overall average. With the assumption that the overall average is close to 33, we can estimate that the winning condition will be around 22 (2/3 * 33). Since each group's average is the result of individual player choices, it's important for your group to cooperate and try to maintain a lower average than the other groups. However, aiming too low may push the overall average down, making the winning condition lower as well.

Based on these considerations, a good strategy would be to choose a number that is lower than the previously reported averages and closer to the estimated winning condition of 22. Assuming that other players will also adjust their strategies, choosing a number between 20 and 25 would maximize your group's chances of winning.

In conclusion, considering the previous round's results and the estimated winning condition of 22, reporting a number between 20 and 25 would be a good strategic choice for this round. The specific number can be determined by communicating with your group members and agreeing on a collective decision. However, as a representative value, 22.5 would be a reasonable choice for this round.

---

**Prompt 8**

You are playing a multiplayer game.

# Game Rule
1. Each player reports a real number between 0 and 100, inclusive.
2. The winner will be the player whose number is the closest to 5 plus 1/2 of the average of all reported numbers.

Think step by step and then report your number.

---

**Experimental Observations**    In the variant game where we change the Nash equilibrium to 10, we observe that agents can give a good understanding of the game and correctly calculate the Nash equilibrium, as shown in Example 16, indicating their powerful reasoning ability.

**Typical Mistakes in Agent Behaviors**    We summarize some typical mistakes made by agents, including simple calculation errors, as shown in Example 17, and making decisions that are inconsistent with the considerations, as shown in Example 18. Although agents might make mistakes, they exhibit powerful reasoning abilities and show great potential for use in simulations.

## H.2   IMPACT OF TEMPERATURE

The temperature serves as a hyperparameter for controlling the diversity of LLM generation. In this subsection, we set up 1,000 agents equipped with Qwen2-72B and Prompt 2, and vary the temperature within the range of $\{0.05, 0.25, 0.50, 1.00\}$. The experimental results are presented in Fig. 21, from which we can observe that as the temperature increases, the standard deviation rises from 6.50 to 8.20 while the average number shows only slight changes. These findings indicate that the generation temperature can impact the distributions of reported numbers, but may not significantly alter the overall average when the number of participating agents is large.

---

**Example 14: Behaviors of an agent in group 2 (Qwen2-72B)**

Given the outcomes of the previous round, we can derive some implications for the next round's strategy:

1. **Past Rounds' Outcomes**:
- 2/3 of the average for the previous round was 22.50.
- Group 1's average was 39.36.
- Group 2's average was 21.14.
- Group 3's average was 40.75.
- Group 2 was the closest, but the averages indicate that players were still reporting numbers relatively higher than the target of 22.50.

2. **Adapt Strategy**:
- If players adjust their strategies to be closer to the target of 22.50 for the next round, we can expect the average to be lower due to a general understanding that numbers should be lower. - Players might anticipate that the average number from all groups could be around 25 to 30 (slightly higher than the target, assuming some overshooting), making the target number approximately 16 to 20.

3. **Further Iterative Thinking**:
- Rational players will likely think ahead, aiming for the new target based on expected adjustment. If everyone thinks this way, the reported numbers may move in a range close to the expected target numbers for this next round.
- Considering the potential for overshooting and the need for adjustment based on iterative thinking, reporting a number that's close to—or slightly below—the expected target could be strategic.

**Decision-Making**:
Based on the above analysis, since the target for this round might be around 18 to 20 (considering previous oversights and adaptive strategies), aiming to report a number in this range would be rational, expecting that other players may also modify their numbers toward this target if it has been acknowledged.

**Final Reported Number**:
Given the previous outcomes and expected adjustments, I would propose reporting 19 for this round. This number is centered in the expected adjusted target range (18-20), taking into account both the iterative thinking and the learning effect from past rounds, striving for a balance between shifting too low (potentially underscoring the target) and not shifting enough (potentially missing the target due to over-adjustment by others). Reporting 19 ensures a strategic middle-ground accounting for possible moves by other players.

---

**Example 15: Behaviors of an agent in group 3 (Llama3-70B)**

New information!

Let's analyze the previous round and think step by step for the next round:

1. The 2/3 of the average was 22.50, which means the overall average was 33.75.
2. The numbers reported by groups are Group 1: 39.36, Group 2: 21.14, and Group 3: 40.75.
3. My group (Group 3) reported an average of 40.75, which was higher than the overall average.
4. Since I reported 64 in the previous round, I contributed to my group's high average.
5. For the next round, I should consider the fact that groups 1 and 2 might adjust their numbers based on the previous round's results.
6. Group 1 reported a relatively high average, so they might try to lower their average to get closer to the 2/3 mark.
7. Group 2 reported a relatively low average, so they might try to increase their average to get closer to the 2/3 mark.
8. My group (Group 3) should aim to report a number that's closer to the 2/3 mark, considering the potential adjustments from groups 1 and 2.
9. Assuming groups 1 and 2 adjust their numbers, the overall average might shift downwards.
10. To anticipate this, I'll report a number that's slightly lower than the 2/3 mark, which is around 22.50.

Considering these factors, my reported number for the next round is:

**20**

---

## H.3 PLAYING AS A SEVEN-YEAR-OLD CHILD

In the simulation involving agents with diverse background settings (see Sec. 4.4), we attempt to assign an agent with an extreme role: playing as a seven-year-old child in the game. From the generation of this agent, as shown in Example 19, we can see that the agent's behaviors do not align with those expected of a seven-year-old child, as they demonstrate abilities such as performing calculations and engaging in multi-step reasoning. These findings highlight

---

**Example 16: Agent behavior: correctly calculating the Nash equilibrium**

To determine the best number to report in this scenario, we have to consider what the optimal strategy would be given that other players are also trying to win under the same rules. This leads us into a game of guessing what everyone else will guess. Let's break down the steps:
1. **Assume everyone reports randomly between 0 and 100**: In this case, the average would tend towards 50, and the optimal play would be 5 + 1/2 * 50 = 30. However, most people are aware of the game theory behind these situations, and they won't play optimally if everyone assumes the average is 50.
2. **Adjust for strategic play**: If everyone assumes that the optimal play is 30 (from step 1), and they all play 30, then the new average is 30. Therefore, the optimal play becomes 5 + 1/2 * 30 = 20. However, if everyone knows this, the optimal play would shift again.
3. **Iterate to find the point of convergence**: Let's assume that after a few rounds of reasoning, players would converge on a number where further deduction does not significantly change their strategy. This point is often known in game theory as the "Nash equilibrium."
To estimate this equilibrium, let's assume that after several rounds of reasoning, players might settle around a number we can estimate through iteration. Simplifying for ease of understanding:
- **First guess (G1):** 5 + 1/2 * 50 = 30
- **Second guess (G2):** 5 + 1/2 * 30 = 20
- **Third guess (G3):** 5 + 1/2 * 20 = 15
- **Fourth guess (G4):** 5 + 1/2 * 15 = 12.5
- **Fifth guess (G5):** 5 + 1/2 * 12.5 = 11.25
- **Sixth guess (G6):** 5 + 1/2 * 11.25 = 10.625
  Given these calculations, the numbers are converging. After a certain number of iterations, the number will stabilize. The key realization is that the strategy relies on predicting the collective behavior of others, who are also trying to predict others' behavior. This leads to a pattern where estimated numbers decrease with each iteration until they stabilize around a specific value, considering the game theoretic approach.
However, it is difficult to know how many rounds of reasoning (iterations) other players are completing, so one might need to make an educated guess based on these iterations. Given that the numbers stabilize in the mid- to low-single digits, a strategic guess might be around 10, considering the patterns of logical reasoning discussed.
Therefore, an educated and strategic guess, in hopes of anticipating others' similar reasoning, might be **10**.

---

**Example 17: Agent behavior: calculation errors**

To approach this game optimally, we need to consider the behavior of other players and how they might think. Let's break it down step by step:

1. **Assume Other Players Act Rationality**: If all players think rationally, they will consider how others might guess, leading to a recursive thought process. We can assume that the first level of thinking is that everyone picks randomly between 0 and 100. The average of such random numbers would also fall within this range, and 51/100 of this average is expected to be around 51 (considering a uniform distribution).

2. **Second-Level Thinking**: If you assume that everyone thinks the average will be around 50 and they go for 51, the new average might shift. If everyone picks 51, the new average would indeed be 51, and 51/100 of this average would still be 51. But with our current game rule, this iterative thinking shouldn't lead to oscillations beyond this point because we are targeting a static multiplier of the average.

3. **Accounting for the Multiplier**: The multiplier, 51/100, makes this interesting because it's slightly more than half. If everyone picks 51, it's already the optimal pick considering others at that point will likely pick similar. Changing the pick in anticipation of everyone adjusting in a certain direction (like picking lower because of the 51/100 factor) might not necessarily benefit you unless there's a collective and predictable shift in thinking.

4. **Avoiding the Lowest and Highest Percentiles**: Picking 0 or 100 could be risky because even if you guess that others will pick high in anticipation of the multiplier effect, these would generally be outliers. Given the multiplier is slightly more than half, it's unlikely that the average will be pulled too high or too low.

Given this analysis, considering the slight uncertainty in how other players might think and adjust, the safest bet would be to stay close to where the effect of the multiplier is directly considered. Therefore, **I will report the number 51**. This assumes others might also gravitate around this number, and it balances well against the 51/100 multiplier applied to the average.

---

the importance for users to evaluate the model's capabilities for playing certain characters and to carefully design the background settings before conducting simulations. Although various background settings can be assigned to agents, they may not exhibit consistent behaviors in some extreme cases.

---

**Example 18: Agent behavior: making decisions that are inconsistent with the considerations**

To approach this game strategically, let's break it down step by step.

First, we know that the target number is not static; it depends on the average of all reported numbers. The formula for the target number is 5 + 1/2 * average.

If we assume everyone else is rational and is also trying to win (which usually is a good assumption in economic games and is called "common knowledge of rationality"), we can hypothesize that they'll also attempt to guess the target number.

If we assume that the initial expectation is that everyone will report a random number between 0 and 100, the average should be 50 (if the numbers are uniformly distributed). Therefore, the target number, using the provided formula, would be 5 + 1/2 * 50 = 5 + 25 = 30.

However, if everyone thinks this way, then the new average would be 30, making the new target 5 + 1/2 * 30 = 20. Yet, if everyone thinks that everyone thinks this way, the next target would be 5 + 1/2 * 20 = 15... and so on.

In game theory, this concept is known as iterated expectations or guess-the-number games. It typically leads to a series of lower expectations each iteration until it reaches a stable equilibrium point where further iteration changes no more. In guess-the-number games where the target is a fraction of the average, that equilibrium is usually the lower boundary, in this case, 0, since the formula can only reduce the number, not increase it again once it reaches the lower boundary.

So, if everyone theoretically iterates their thinking to its conclusion and reaches the point where the target stops changing, they should all guess 0. However, that's assuming perfect common knowledge of rationality and infinite iteration which is unrealistic in a real game. People may not fully converge to 0 due to common knowledge failing or because they do not iterate their iteration to the full extent.

Given that, while guessing 0 might be an interesting strategic play, considering human behavior, a number very close to 0 but not exactly 0 might actually be a smart choice, anticipating that others might also guess a small positive number.

However, in a simplification and assuming everyone is eventually attracted to the theoretical equilibrium, I will predict 0, understanding the likely outcome based on common strategy in these types of games is closer to but slightly above 0.

Here is my official number: 0.

Please note: this is a simplification of game theory concepts. In real-world games, people don't always behave rationally or predictably, causing actual outcomes to vary. The goal of game theory is not to predicted exact outcomes but to understand how incentives shape individual decisions.

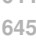

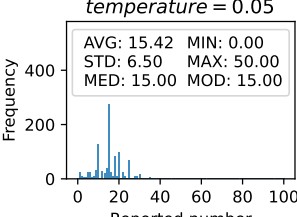 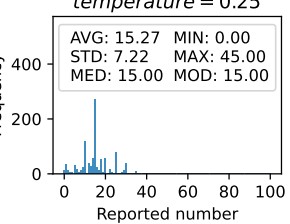 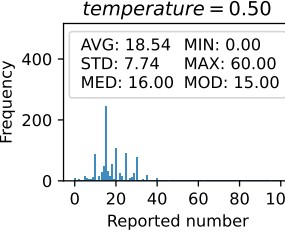 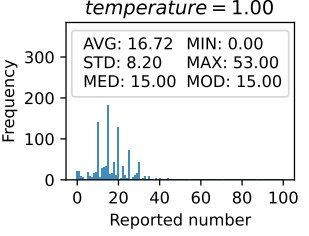

Figure 21: The distributions of reported numbers when using different temperatures in LLMs.

---

**Example 19: Agent behavior: playing as a seven-year-old child**

Step 1: Understanding the Game
The main goal of the game is to guess the closest number to 2/3 of the average number all players have submitted. Considering that the average number might swing wide due to other players' submissions, it's usually wise to choose a number less than 66.67 (which is 2/3 of 100), but also taking into account that other players might think similarly, thus bringing down the average.

Step 2: Applying Logical Reasoning
Being a young student, my logical framework may not be as complex as that of adults, but I can still think through the process. Let's assume everyone chooses a number between 0 and 66.67 because 2/3 of 100 is 66.67. If everyone's thinking this way, the winning number might actually be closer to 2/3 of 66.67, since that's what most people are guessing close to.

Step 3: Taking a Reasoned Guess
Now, to guess the actual number, I have to think of what 2/3 of 66.67 is, which comes out to be approximately 44.44. But since other players might be thinking this too, I should guess below 44.44.

Considering that 7-year-olds would generally not calculate the arithmetic exactly, I would instinctively guess a little lower than the final calculated value. Being young might mean I'd pick a number that is round or easy to think of. So, my final guess would be 40.

Reported Number: 40

