# OpenReview forum: "Very Large-Scale Multi-Agent Simulation with LLM-Powered Agents"
_ICLR.cc/2025/Conference — Submitted to ICLR 2025_

### Official Review · Reviewer_yfvq · 2024-10-19

**Soundness:** 1
**Presentation:** 4
**Contribution:** 1
**Rating:** 3
**Confidence:** 5

**Summary:**

The paper first proposes an actor-based distributed mechanisim to implement very large-scale multi-agent simualtion with LLM powered agent. And it also shows several components to make it easy-to-use including the `to_dist` function, environments, configurable tools, Agent-Manager, etc.
Based on the system, the paper conducts several experiments about *guess the 2/3 of the average* game to show the performance of the system, the game result of different LLMs and different prompts. Some deeper discussion of the LLMs' outputs with non-trivial settings are shown.

**Strengths:**

1. The proposed actor-based distributed mechanism is reasonable and effective for realizing large-scale distributed computation acceleration for multi-agent simulation tasks.
2. The proposed toolchains including the configurable tool, the background generation pipeline, Agent-Manager can be very effective in saving the workload of researchers and making the system easy to use.
3. The writing is very fluid and easy to understand, with no grammatical problems.

**Weaknesses:**

# Summary

Overall, I am very confused about the topic of this paper.

Based on the keywords and the abstract of this paper, it seems that it wants to introduce a software framework/system to support very large-scale multi-agent (LLM Agent) simulation.
But the paper devotes only a relatively small amount of space to its key technologies (Sec 3.1 and Sec 3.2).
Plenty of content (Sec 4.3, Sec 4.4, Sec 4.5 and Sec 4.6) is discussing how different LLMs perform on the *guess 2/3* game.

Besides, the key contribution (actor-based distribution mechanism) shown in the abstract has been mentioned in the literature (AgentScope) cited herein [1] and both papers have similar figures (Fig. 1 in the paper vs Fig. 8 in [1]).

# Weaknesses

I will list below, point by point, what I consider to be the weaknesses of this paper:
1. What is the relationship between the paper and AgentScope[1] published in Arxiv? Is this paper a follow-up to AgentScope or its official submission version? If it is the former, the main contribution of this article will not be valid; if it is the latter, why cite AgentScope in the paper.
2. In Sec. 3.2, what dose it means *"the environment is expected to meet the following requirements: ... (d) Multiple environments"*? Do the two environments refer to the same thing?
3. In Sec. 3.2, there is no details about *listeners*. But this is an important concept/component for realizing the bi-directional interactions between the agents and the environments.
4. The *guess the 2/3 of the average* game is too easy for distributed multi-agent simulation. There is no dependency among any agents. The game is a simple map-reduce task in which all agents report numbers based on the previous round and aggregate for averaging. Testing in this setting cannot evaluate the real performance of the proposed DAG-based/actor-based distributed mechanism. The finding*"Increasing the number of devices can proportionally reduce the simulation running time."* mentioned in Sec. 4.2 should be attributed mainly to the setting with no dependencies.
5. The related discussion (Sec. 4.3, part of Sec. 4.4, Sec. 4.5, and Sec. 4.6) about the behavior of different LLMs on this game seems irrelevant to the topic of this paper.

# Reference

[1] Dawei Gao, Zitao Li, Xuchen Pan, Weirui Kuang, Zhijian Ma, Bingchen Qian, Fei Wei, Wenhao Zhang, Yuexiang Xie, Daoyuan Chen, Liuyi Yao, Hongyi Peng, Zeyu Zhang, Lin Zhu, Chen Cheng, Hongzhu Shi, Yaliang Li, Bolin Ding, and Jingren Zhou. Agentscope: A flexible yet robust multi-agent platform. CoRR, abs/2402.14034, 2024.

**Questions:**

My questions are as follows:
1. What is the paper's main topic and key contributions?
2. What is *listener* and how to use it or implement it?
3. Why the number of agents is not consistent across experiments? Sec. 4.2 (i): 1M agents; Sec. 4.2 (iii): 10,000 agents; Sec. 4.4: 200*NUM_GROUPS agents; Sec. 4.5: 500*3 agents.
4. What is the performance of the system in a simulation setting with more complex dependencies?
5. Why does it take 8.6 hours to run 1M sleeps 1 second in asynchronous mode mentioned in Sec. 4.2 (ii).
6. How to distinguish between LLM inference time, agent processing time, and additional overhead due to communication? What are their ratios? This is because when we talk about distributed computing, the main cost of horizontal scaling of computing power is the additional overhead due to communication.

**Details Of Ethics Concerns:**

Similar figures between the paper and its cited paper.

Figure 1 in the paper vs Figure 8 in [1] (in the paper, the cite is shown as `(Gao et al., 2024)`)

[1] Dawei Gao, Zitao Li, Xuchen Pan, Weirui Kuang, Zhijian Ma, Bingchen Qian, Fei Wei, Wenhao Zhang, Yuexiang Xie, Daoyuan Chen, Liuyi Yao, Hongyi Peng, Zeyu Zhang, Lin Zhu, Chen Cheng, Hongzhu Shi, Yaliang Li, Bolin Ding, and Jingren Zhou. Agentscope: A flexible yet robust multi-agent platform. CoRR, abs/2402.14034, 2024.

---

> ### Author Response · Authors · 2024-12-02
>
> Thank you very much for the time and effort you put into reviewing our paper!
> We have carefully considered each of your helpful suggestions and will include the corresponding revisions and enhancements in a new version of our paper. For more detailed responses, please see below.
>
> ---
>
> **Responses to W1**:  Thank you for your comments. To promote the essential functionalities requisite for agent-based simulations, we have developed a series of novel features and components that enhance AgentScope to support large-scale simulations involving multiple agents.
>
> **Responses to W2**: Thank you for your comments. Here we indicate that, the environment module in the framework is expected to support the situation that needs multiple environments (such as discussions in multiple groups). We will make these descriptions more clear.
>
> **Responses to W3 \& Q2**:  Thank you for your suggestions regarding the listeners. We are pleased to add more details about the listeners as it is an important component for realizing the bi-directional interactions between the agents and the environments.
>
> **Responses to W4 \& Q4**: Thank you for your suggestions. We will conduct explorations on more complex multi-agent interaction patterns.
>
> **Responses to W5 \& Q1**: Our framework is designed to effectively and efficiently support very large-scale simulations, enabling LLM-empowered agents to perform considerations and reasoning behaviors consistent with those observed in social simulations. These experiments and discussions in Section 4.3-4.6 are provided to show how agents perform considerations and reasoning behaviors in agent-based simulations, demonstrating the great potential of LLM-empowered agents in simulations built upon a scalable and flexible platform, which is within the scope of this paper.
>
> **Responses to Q3**: Thank you for your comments. The number of agents might be set to different values in different experiments due to the different purposes of the conducted experiments. For example, the experiments in Section 4.2 are conducted to show the maximum number of agents that the framework can support within a certain number of devices, while other experiments are designed to show different reasoning behaviors of agents in different scenarios.
>
> **Responses to Q5**: Thank you for your comments.
> Although API calls use sleep(1) for simulation, other aspects such as prompt construction, parsing of output results, and interactions between agents will still consume CPU resources, which would be limited by GIL.
>
> **Responses to Q6**: From the experimental results in Section 4.2, we can observe that the time taken for LLM API calls is the primary bottleneck, accounting for more than 99% of the total experiment duration. The cost of agent processing and communication is less than 1%. In the studied agent scenarios, different from traditional distributed computing, the communications between agents are not a bottleneck as they only exchange simple strings of relatively short lengths (not exceeding 1,000 tokens).
>
> ---
>
> Thank you again for your detailed comments and helpful suggestions!

---

> > ### Comment · Reviewer_yfvq · 2024-12-03
> >
> > Thank you very much for your reply, but there doesn't seem to be enough time right now to accomplish the improvements you promised.

---

### Official Review · Reviewer_zhRB · 2024-10-27

**Soundness:** 3
**Presentation:** 1
**Contribution:** 2
**Rating:** 3
**Confidence:** 4

**Summary:**

Although there is no clear section or subsection for listing the contributions, in summary, a distributed actor-based simulation framework is presented that enables parallel execution of agents using the already established techniques. Then the paper discusses the behavior of different LLMs by analyzing one particular simulation.

Although the discussion about the LLMs and their behavior in different settings is interesting, the paper lacks the overall purpose. At the end of the introduction section, some contributions are mentioned, but all of them are not treated thoroughly. It would have been better to focus on just one or two things and then draw concrete conclusions.

What has been achieved is not mentioned clearly. There should have been a bulleted list of contributions rather than a paragraph that mixes many different things. From lines 57 to 83 the following can be deduced.

1 - The Authors propose a distributed mechanism based on the actor model that allows agent-level parallel execution and automatic workflow conversion.
2 -  Users can convert a linear simulation into a distributed one by calling a simple function to_dist().
3 - Then the authors mention some functionalities that are present in almost all agent-based frameworks, like inter-agent and agent-environment interactions, user-defined functions, and their query mechanisms.
4 - Authors provide an automatic configuration tool and an automatic background generation pipeline.
5 - Then there is a component named agent manager that enables the monitoring and organization of the agents
6 - Using the framework they have developed they simulated the game of "guess 2/3 of the average"
7 - The authors emphasize at the end of section 1 that "these experimental results confirm the feasibility and great potential of conducting large-scale agent-based simulations with LLM-powered agents."

Please consider creating a separate section for "contributions" and enumerate them as a bulleted list. You can place this section at the end of the introduction section.

Here are the issues with all of the above contributions
1 - The authors do not propose the actor model, and agent-level parallelism using the RPC calls. What they mean by workflow conversion is not clear. Please clarify what specific innovations or improvements their approach makes to existing actor-model and RPC-based parallelism techniques.
2 - How this function works is unclear. For example, how do the two stages mentioned in the section "Automatic workflow conversion" actually achieve their target? The description is very brief and difficult to analyze for correctness. In fact, this seems like the major contribution but it lacks the details. It only spans 9 lines.  Please add more details on what 'workflow conversion' entails in this context. Also, provide more detailed explanations of how the two stages work and achieve their targets. This would help readers better understand and evaluate this key contribution.
3 - These functionalities are common among agent-based frameworks.
4 - The automatic configuration tool is the widely used distribution template (line 215)
5 - The authors have only dedicated 10 lines to this topic referring to appendix B.4 which is simply a screenshot of the proposed manager without any clarity of identification of components. Please expand the section about Agent Manager. Also, provide clearer labels or annotations in Appendix B.4 to highlight the key components and functionalities of the interface.
6 - This is not significant as the results ought to be similar to previous results. Although from this example some insights for the LLMs could have been deduced, because they are not deduced systematically, they are lost in another discussion. Probably if the author had focused only on this aspect and had compared more than one example this could be considered a valuable contribution.
7 - This confirmation is widely supported by many researchers so suggesting this as a contribution probably is an overstatement.

**Strengths:**

The analysis and comparison of different LLMs in different settings is interesting but this seems to be left in the middle. There is only one example discussed, and writers admit that the example seems to already present in the knowledge of LLM which makes the comparison more skeptical as it is not clear which model knew about the problem to which extent. This also raises the question why not create a new problem and run experiments on those. Only one example is always too few examples. If you have to pick few examples then they must be chosen on sound basis. In the article there is no argument presented in favor of choosing the "guess 2/3 of average" example.

**Weaknesses:**

- If the authors wanted to focus on making the simulation framework more scalable and efficient then should have only focused on this task and should have presented the novelty of their idea and improvement by comparing other previously developed frameworks.
- The function to_dist is a mystery, it is not clear how it work when there is a lot to be done in terms of creating proxies and RPC calls. Example in Figure 1 does not answer the questions.
- Line 231-232 : "To tackle this, we incorporate advanced forms of agent management and monitoring, named Agent-Manager", this is a strange statement at this level. Which "advanced forms" are used here, are not clear.
- Line 514 refers to Figure 9, but the figure does not support the claim in my opinion.

It seems either the appendices are numbered wrong or they are missing the information that was intended to be there for example.
	- Line 517 says "findings are further confirmed by observations of individual-level behaviors shown in Appendix H.1." but appendix H.1 seems to contain the same information that was already mentioned.
	- Line 234-236 says," These servers are responsible for managing the lifecycle of distributed
	agents and synchronizing their information to a web-based visual interface, which is provided for a comprehensive
overview of all registered servers and all deployed agents on different devices, as shown in Appendix B.4" but the Appendix B.4 is just a screenshot of a GUI without highlighting different areas that represent the server’s identity, IP address, running status, and utilization of computing resources.

**Questions:**

The review section already includes suggestions to focus on one task and draw concrete conclusions after performing the complete scientific process. Do not create the article a mixture of many unproven jestures.

---

> ### Author Response · Authors · 2024-12-02
>
> Thank you very much for the time and effort you put into reviewing our paper!
> We have carefully considered each of your helpful suggestions and will include the corresponding revisions and enhancements in a new version of our paper. For more detailed responses, please see below.
>
> ---
>
> **Responses to W1**: Thank you for your suggestions! The comparisons summarized in Figure 2, along with the discussions provided in Section 4.2, demonstrate the superior performance of what we proposed in this paper compared to those commonly used in existing multi-agent frameworks. We would like to provide further details and experiments to highlight the technical contributions of our study.
>
> **Responses to W2**: Thank you for your suggestions. We will add more details of the function to_dist to make it more clear.
>
> **Responses to W3**: Thank you for your comments. We will polish the description of Agent-Manager to make it more clear in the new version of our paper.
>
>
> **Responses to W4**: Thank you very much for your comments regarding the Appendix. We will carefully review and polish the contents in the Appendix.
>
> ---
>
> Thank you again for your detailed comments and helpful suggestions!

---

### Official Review · Reviewer_KCn2 · 2024-11-03

**Soundness:** 1
**Presentation:** 2
**Contribution:** 1
**Rating:** 3
**Confidence:** 4

**Summary:**

The paper introduces a framework for very large-scale multi-agent simulations powered by LLMs. The motivation for using LLMs is to create diverse, adaptive, and realistic agent behaviours across massive agent populations. The proposed system employs an what is called an actor-based distributed mechanism to handle parallel execution of up to a million agents, a kind of message passing approach. The idea is that this supports scalable simulations while making good use of resources. It includes tools for generating diverse agent profiles, such as varying age, occupation, and education, to reflect real-world demographic diversity. Additionally, a web-based interface is provided for efficient configuration, monitoring, and management of distributed agents across devices.

As a demonstration, the framework applies the "guess 2/3 of the average" game, where agents engage in iterative rounds and adjust strategies, showcasing the platform’s ability to simulate complex, collective decision-making. With prompt-driven LLMs, agents can react in contextually nuanced ways, making this setup particularly suitable for studying collective dynamics and interactions among heterogeneous populations. The platform appears to be intended for applications in fields such as social science, behavioural economics, and multi-agent systems research.

The paper has a strong focus on agent dynamics and scalability, its core contributions are in multi-agent research / management of distributed systems.

**Strengths:**

The proposed system appears to be able to handle a large number of agents; the design and scalability seem to be strong as far as I can tell, to be able to cope with a large number of agents. The work also covers agent diversity well.

Some effort appears to have gone into the engineering and useability of the system. The paper also has an extensive appendix with additional insights (I haven't read all of it, but it appears to contain a good amount of information).

**Weaknesses:**

The main weakness is that the paper does not appear to be a good fit for ICLR - it is making use of machine learning models but it's main contributions are outside of machine learning. The lack of a genuine machine learning contribution is my main reason to recommend rejection. Apart from the poor fit, there are multiple critical issues that I feel should be addressed to make the work more accessible.

The paper isn't written in a way that makes it easy to see what is the practical application or research focus of the work, and feels like it has been written for another audience. The application areas of the simulations dicsussed in the paper can be broadly inferred, but apart from the "guess the 2/3" game I am a bit lost with the direction. The paper starts talking about simulations / simulation platforms without providing a meaningful context what kind of simulations the work is considering, or what questions the simulations are supposed to answer.
Without that direction, it is remains unclear to me also why LLMs are useful / necessary / a good idea for this work, or if a simpler model might have been possible. Maybe there would be room for a comparison that helps the reader understand this better. Given the limited amount of comparison to simpler approaches, coming up with a meaningful baseline would be helpful.

The work discusses scalability and an actor-based distributed mechanism that allows for parallel execution of agents, but I would have expected specifics on how to handle the high resource demands of LLMs, or if it is just a matter of running everything in parallel (what are the contributions in this area).

I understand the "guess the 2/3 average" is an example experiment, but to show that the system is capable of solving complex scenarios more complex evaluations would be helpful. Using LLms for this seems a bit overengineered and computationally demanding, though I may have missed a point. For meaningful benchmarking of the scalability, I would have expected a broader range of evaluations.

**Questions:**

- Can you clarify the specific practical applications or research questions your framework is designed to address?
- Why are LLMs necessary for these simulations, and have you considered simpler models or compared performance?
- How do you handle the computational requirements of running large numbers of LLM agents? Are there specific load-balancing or resource management strategies?
- What kinds of behaviours the LLMs are generating? Ie how are the responses used exactly, for direct decision-making, or how do they determine agent behaviour?

---

> ### Author Response · Authors · 2024-12-02
>
> Thank you very much for the time and effort you put into reviewing our paper!
> We have carefully considered each of your helpful suggestions and will include the corresponding revisions and enhancements in a new version of our paper. For more detailed responses, please see below.
>
> ---
>
> **Responses to Q1**: Thank you for your comments. Our framework is designed to effectively and efficiently support very large-scale simulations, enabling LLM-empowered agents to perform considerations and reasoning behaviors consistent with those observed in social simulations. With the incorporation of LLM-empowered agents, simulations are expected to become more interactive, adaptive, and realistic, while requiring substantially fewer human efforts.
>
> **Responses to Q2**: Thank you for your comments. We use LLM-empowered agents due to their impressive capabilities in natural language understanding, processing, and generating, which can effectively satisfy the diverse requirements brought by conducting various types of simulations. In this study, instead of providing support for one or several specific types of simulations, we focus on designing functionalities and improvements on multi-agent systems from a more general perspective, to demonstrate the great potential of LLM-empowered agents in simulations built upon a scalable and flexible platform.
>
> **Responses to Q3**: Thank you for your comments. In this study, the API requests from agents are evenly distributed across different model instances, and each agent is associated with a specific model inference service instance for load-balancing.
>
> **Responses to Q4**: Thank you for your comments. Please refer to the Appendix for detailed examples of agent behaviors.
>
> ---
>
> Thank you again for your detailed comments and helpful suggestions!

---

### Official Review · Reviewer_n8xh · 2024-11-03

**Soundness:** 3
**Presentation:** 3
**Contribution:** 2
**Rating:** 3
**Confidence:** 4

**Summary:**

This paper presents a framework for large-scale multi-agent simulations powered by Large Language Models. The authors propose several key components including an actor-based distributed mechanism for parallel execution, flexible environment support for agent interactions, configurable tools for generating diverse agent backgrounds, and a web-based interface for agent management. The framework demonstrates the capability to support simulations involving up to 1 million agents across multiple devices. The authors conduct comprehensive experiments using the classic "guess 2/3 of the average" game to evaluate different aspects of the framework, including scalability, efficiency, agent diversity, and behavioral patterns.

**Strengths:**

S1. The system architecture is well-designed with a clear separation of components and modular design.

S2. The scaling-up simulation size problem is an important challenge in multi-agent systems research.

S3. Extensive experimental analysis across different LLMs, prompts, and agent configurations.

**Weaknesses:**

W1. While the paper claims to support both inter-agent and agent-environment interactions, the experiments primarily demonstrate agent-environment interactions through the game setting, lacking direct agent-to-agent communication scenarios. There is limited evaluation of inter-agent interactions.

W2. The extensive ablation studies concentrate on comparing different LLMs and prompts in the 2/3 game, rather than evaluating the scalability and efficiency of the proposed distributed mechanism and other architectural components, which in my opinion is more important.

W3. The technical contributions are limited in several aspects, the actor-based distributed mechanism follows common patterns in distributed computing frameworks without significant innovations, the environment module uses standard multi-agent system design patterns, and the agent background generation and web interface components are useful but not technically novel.

W4. The parallel processing approach may face practical limitations with GPU memory constraints or API rate limits when using external LLM services. In addition, it is difficult to perform parallel computation because it is hard to determine in advance how the agents will interact with each other.

**Questions:**

Following the above discussion:

Q1. Is it possible to evaluate scenarios with direct agent-to-agent communications rather than just environment-mediated interactions?

Q2. What are the unique advantages of the proposed distributed mechanism compared to existing distributed computing frameworks? Could you provide ablation studies focusing on the performance gains from the architectural components?

Q3. How does the system handle resource constraints (GPU memory, API rate limits) in real-world deployments?

Q4. Could it be considered to evaluate more complex multi-agent interaction patterns that go beyond the simple game theory example?

Q5. Is it possible to quantify the performance improvements and resource utilization of your distributed architecture compared to baseline implementations?

---

> ### Author Response · Authors · 2024-12-02
>
> Thank you very much for the time and effort you put into reviewing our paper!
> We have carefully considered each of your helpful suggestions and will include the corresponding revisions and enhancements in a new version of our paper. For more detailed responses, please see below.
>
> ---
>
> **Responses to W1 \& Q1**: Thank you for your suggestions regarding the agent-to-agent communication scenarios. The group-level simulation shown in Section 4.5 is a simplified experiment in an agent-to-agent communication scenario. We agree that conducting additional experiments in agent-to-agent communication scenarios can be a great idea to better demonstrate the effectiveness of the proposed framework in supporting inter-agent interactions.
>
> **Responses to W2**: Thank you for your suggestions. These ablation studies are conducted to show how agents perform considerations and reasoning behaviors in agent-based simulations, which is within the scope of this paper. Meanwhile, we are glad to include more experiments to demonstrate the scalability and efficiency of the proposed framework.
>
> **Responses to W3**: Thank you for your comments. Applying these techniques to agent-based simulations requires non-trivial adaption and improvements, especially considering the scalability, usability, and flexibility of the proposed framework. We will provide further details and experiments to highlight the technical contributions of our study.
>
> **Responses to W4 \& Q3**: Thank you for your comments. The proposed framework utilizes advanced inference engines (such as vLLM) to effectively address the issue of GPU resource utilization, and adopts the Actor model to handle parallel execution of dynamic graphs.
>
> **Responses to Q2**: Thank you for your comments. The focus of this study is to enhance the scalability, usability, and flexibility of the multi-agent systems to support very large-scale simulations. The comparisons summarized in Figure 2, along with the discussions provided in Section 4.2, demonstrate the superior performance of what we proposed in this paper compared to those commonly used in existing multi-agent frameworks.
>
> **Responses to Q4**: Thank you for your suggestions. We will conduct explorations on more complex multi-agent interaction patterns.
>
> **Responses to Q5**: Thank you for your suggestions. We will add more quantification results to show the performance improvements and resource utilization of the proposed framework.
>
> ---
>
> Thank you again for your detailed comments and helpful suggestions!

---

### Meta-Review · Area_Chair_UXEQ · 2024-12-29

**Metareview:**

The paper proposes LLM-based large-scale multi-agent simulation framework powered, featuring an actor-based distributed mechanism for parallel execution, flexible environment support, and tools for creating diverse agent backgrounds. The framework enables simulations with up to 1 million agents across multiple devices and provides a web-based interface for monitoring and management, with the goal of supporting complex, interactive, and realistic simulations with fewer human efforts.

The reviewers acknowledge the potential of the framework for large-scale multi-agent simulation, but highlight the need for a clearer focus on machine learning contributions, more detailed technical explanations, more meaningful experiments, and a better presentation of the work.

**Additional Comments On Reviewer Discussion:**

The reviewers appreciate the authors' responses and commitment to make the changes, but the sentiment is that the changes are so substantial to be made without additional peer review.

---

### Decision · Program_Chairs · 2025-01-22

Reject